# TEADs, Yap, Taz, Vgll4s transcription factors control the establishment of Left-Right asymmetry in zebrafish

Jonathan Fillatre[1], Jean-Daniel Fauny[2,3], Jasmine Alexandra Fels[1], Cheng Li[4], Mary Goll[4], Christine Thisse[1,2], Bernard Thisse[1,2]*

[1]Department of Cell Biology, University of Virginia, Charlottesville, United States; [2]Institut de Génétique et de Biologie Moléculaire et Cellulaire, CNRS/INSERM/ Université de Strasbourg, Illkirch-Graffenstaden, France; [3]Institut de Biologie Moléculaire et Cellulaire, Strasbourg, France; [4]Department of Genetics, University of Georgia, Athens, United States

**Abstract** In many vertebrates, establishment of Left-Right (LR) asymmetry results from the activity of a ciliated organ functioning as the LR Organizer (LRO). While regulation of the formation of this structure by major signaling pathways has been described, the transcriptional control of LRO formation is poorly understood. Using the zebrafish model, we show that the transcription factors and cofactors mediating or regulating the transcriptional outcome of the Hippo signaling pathway play a pivotal role in controlling the expression of genes essential to the formation of the LRO including ligands and receptors of signaling pathways involved in this process and most genes required for motile ciliogenesis. Moreover, the transcription cofactor, Vgll4l regulates epigenetic programming in LRO progenitors by controlling the expression of writers and readers of DNA methylation marks. Altogether, our study uncovers a novel and essential role for the transcriptional effectors and regulators of the Hippo pathway in establishing LR asymmetry.

DOI: https://doi.org/10.7554/eLife.45241.001

*For correspondence:
bernardthisse@virginia.edu

**Competing interests:** The authors declare that no competing interests exist.

## Introduction

Formation of organs during embryonic development requires a progressive restriction of lineage potential. This process, controlled by major signaling pathways, is achieved through changes in chromatin and in transcription factor (TF) networks: genes associated with pluripotency are progressively silenced by DNA methylation, histone modifications and chromatin compaction while key TFs selectively activate the expression of tissue specific genes (*Boland et al., 2014*; *Reik, 2007*). Therefore, knowing how the activity of TFs and epigenetic modification of the chromatin control organogenesis in vivo is essential to our understanding of both normal development and diseases. As a model system for organogenesis, we use the formation of the Kupffer's vesicle (KV), the first organ formed in the zebrafish embryo and that functions as the Left-Right Organizer (LRO). The KV is the fish homolog of the ventral node of the mouse, the *Xenopus* gastrocoel roof plate and the notochordal plate in rabbit. This organ is composed of ~50 monociliated cells organized as a hollow sphere with motile cilia facing its lumen. Rotation of these cilia generates a transient counterclockwise fluid flow that directs asymmetric activation of a conserved Nodal signaling pathway that guides asymmetric morphogenesis of developing organs (*Dasgupta and Amack, 2016*). This vesicle derives from a small population of ~20 precursor cells called the dorsal forerunner cells (DFCs), which are specified at the dorsal margin of the embryo at the onset of gastrulation in response to Nodal signaling (*Essner et al., 2005*; *Oteiza et al., 2008*). During gastrulation, DFCs arrange into a cluster that undergoes progressive compaction, followed by a mesenchymal to epithelial transition and

organization of a single rosette. Following rosette formation, the center of this rosette opens to progressively give rise to the lumen of the differentiated KV. Finally, ciliogenesis takes place during the last phases of differentiation of DFCs into the KV. Altogether, the epithelial organization of KV progenitors associated with both luminogenesis and ciliogenesis leads to the formation of a functional LRO (*Matsui and Bessho, 2012*).

The regulation of the organogenesis of the LRO, from the specification of its progenitors to a fully functional KV, is well described and involves the activity of Nodal, FGF, non-canonical Wnt, Notch and Hedgehog signaling pathways (*Matsui and Bessho, 2012*). Conversely, a very limited number of TFs expressed in DFCs has been implicated in this process. Six genes have been identified so far, two Sox TFs: Sox32 and Sox17; two T-box TFs: Tbxta (also known as *notail*) and Tbx16 (also known as *spadetail*) and two TFs required for ciliogenesis: Foxj1a and Rfx2 (*Aamar and Dawid, 2010*; *Amack and Yost, 2004*; *Bisgrove et al., 2012*; *Kikuchi et al., 2001*; *Yu et al., 2008*). It is highly unlikely that these six TFs are the only transcriptional regulators of the developmental program leading to the formation of the LRO. Indeed, in this study, we identified six additional transcription factors (TFs) and/or cofactors (TcoFs) crucial for the formation and function of the KV. Strikingly, although the Hippo signaling pathway was previously identified as a major regulator of tissue growth and organ size (*Johnson and Halder, 2014*; *Zhao et al., 2011*), we discovered that the DNA binding TFs (Tead1a and Tead3a), the TcoFs mediators of the Hippo signaling pathway (Yap and Taz) as well as the TcoFs Vgll4b and Vgll4l (two factors homologous to the mammalian Vgll4 that negatively regulates the activity of Yap and Taz) are upstream regulators of the formation and function of the LRO. These TFs and TcoFs (collectively named Hippo TFs/TcoFs thereafter in the text) control the function of signaling pathways involved in this process as well as the expression of genes essential to the formation and function of a ciliated epithelium with motile cilia. Finally, we identified that Vgll4l controls the expression in LRO progenitors of epigenetic factors, writers (the de novo DNA methyltransferases) and readers (Methyl-CpG binding domain proteins) of DNA methylation marks that we found essential for DFCs proliferation and survival as well as for the formation of motile cilia.

## Results

### Hippo TFs/TcoFs are required for the establishment of LR asymmetry

To identify novel factors involved in the transcriptional regulation of the formation of the LRO we screened the zebrafish gene expression patterns database (http://zfin.org/) for TFs or TcoFs specifically expressed in the KV and/or in its progenitors. By this approach, we identified Vgll4l, a TcoF member of the Vestigial like four family, which is strongly expressed at gastrula stage in LRO progenitors (*Figure 1A*).

Vestigial like family members are TcoFs known to function mainly through the interaction with TEA domain DNA-binding family of transcription factors (TEAD) (*Deng and Fang, 2018*). TEADs are the DNA binding TFs to which the TcoFs that mediate the transcriptional outcome of the Hippo signaling pathway, Yap and Taz (also known as Wwtr1), bind to, to activate expression of their target genes. Interestingly, in various human cancer cell lines, Vgll4 was shown to negatively regulate the transcriptional outcome of Hippo signaling by competing with Yap and Taz for TEADs, therefore inhibiting their function (*Zhang et al., 2014*).

In addition to Yap and Taz, the zebrafish genome codes for three members of the Vgll4 family (Vgll4a, Vgll4b, Vgll4l) and four TEADs (Tead1a, Tead1b, Tead3a, Tead3b). However, only Vgll4l, Vgll4b, Yap, Taz, Tead1a and Tead3a are expressed in KV and/or KV progenitors (*Figure 1A*, *Figure 1—figure supplement 1*). To investigate the function of these TFs and TcoFs in the formation of the LRO we performed general and/or DFC specific (*Amack and Yost, 2004*) knockdown experiments using translation interfering and/or splice interfering morpholinos (MOs) and analyzed the effect of these loss of functions on the establishment of the LR asymmetry of the embryo. Looking at the direction of the heart jog at one day of development (*Figure 1B*) and at the expression of *lefty1* in dorsal diencephalon (*Thisse and Thisse, 1999*) at 20 hr post fertilization (*Figure 1—figure supplement 2*) we found that loss of function of each of these TFs or TcoFs strongly disrupts embryo laterality. Specificity of knockdown phenotypes was demonstrated by their reproducibility using different non overlapping MOs and in rescue experiments through injection of in vitro synthesized, MOs insensitive, mRNAs (*Figure 1B*). The implication of Yap in embryo laterality was further confirmed

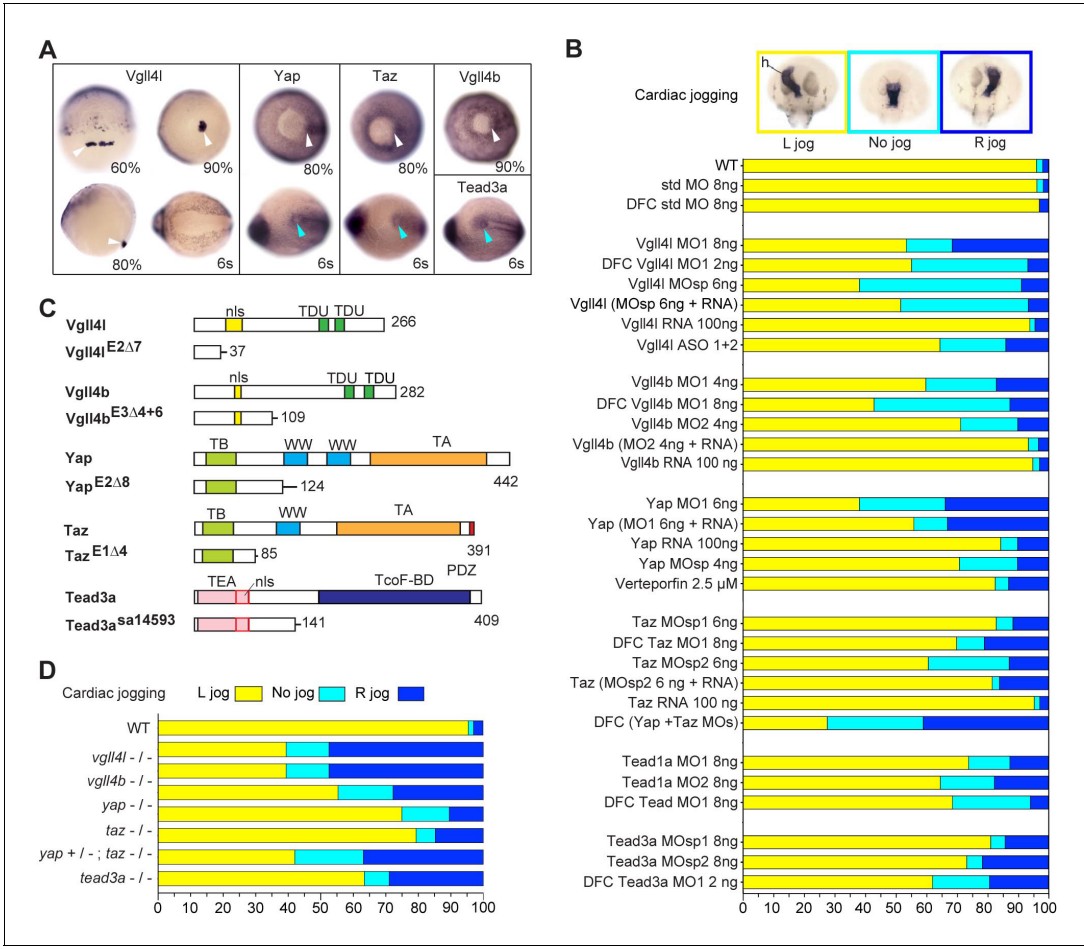

**Figure 1.** Hippo TFs/TcoFs are essential for establishing the left-right asymmetry. (A) Whole-mount in situ hybridization for *vgll4l*, *yap*, *taz*, *vgll4b* and *tead3a* at gastrula and at the 6-somite stage. Vgll4l is expressed in DFCs at 60% (dorsal view), 80% (lateral view) and 90% epiboly (vegetal pole view) but is not expressed in the KV at the 6-somite stage (vegetal pole view). Yap and Taz are shown at 80% epiboly and at the 6-somite stage in vegetal pole view, Vgll4b at gastrula stage and Tead3a at the 6-somite stage in vegetal pole views. Tead1a that is constitutively expressed is not presented. White arrowheads point to DFCs, blue arrowheads point to the KV. (B) Cardiac jogging analyzed at 25 hr post fertilization (hpf). Graphs indicate the percentage of embryos with normal Left jog (L jog - yellow), Right jog (R jog – dark blue) or no jog (light blue), visualized by in situ hybridization (top, h: heart) with *a myosin light chain 7 (myl7)* probed at 25 hpf in: wild-type (WT) embryos; embryos injected with standard (std) MO or with Vgll4l, Vgll4b, Yap, Taz, Yap and Taz, Tead1a or Tead3a MOs; embryos injected with ASO; rescue experiments of morphant phenotypes by injection of MO insensitive RNA; incubation with 2.5 µM of Verteporfin, a Yap inhibitor. For each experiment the name of gene, name and amount of MO and/or RNA injected are indicated on the left. For double Yap/Taz MO-KD, 4 ng Yap MOsp and 4 ng Taz MOsp2 have been injected. DFC 'name of the gene' MO indicates DFC-targeted knockdown experiment (*Wang et al., 2013*). (MO + RNA) stands for rescue experiment of the indicated MO together with 100 ng of the corresponding, MO insensitive, mRNA. (C) Schematic of functional domains present in WT and in Vgll4l, Vgll4b, Yap, Taz and Tead3a mutants. nls: nuclear localization signal, PDZ: PDZ-binding motif, TA: transcription activation domain, TB: TEAD binding domain, TcoF-BD: transcription cofactor binding domain, TEA: DNA-binding TEA/ATTS domain, TDU: TONDU domain, WW: WW domain. Numbers indicate the position of the last amino-acid of each peptide. (D) Laterality defects of homozygous mutant embryos and of embryos homozygous mutant for Taz, heterozygous for Yap, analyzed as described in (B) for their cardiac jogging at 25 hpf. Numerical data for (B) and (D) are provided in *Figure 1—source data 1*.

DOI: https://doi.org/10.7554/eLife.45241.002

The following source data and figure supplements are available for figure 1:

**Source data 1.** Numerical data for *Figure 1B, C, and D*.

DOI: https://doi.org/10.7554/eLife.45241.007

**Figure supplement 1.** Immunodetection of Yap protein in nuclei of DFCs.

DOI: https://doi.org/10.7554/eLife.45241.003

**Figure supplement 2.** Expression of *lefty1* (*lft1*) at 20 hpf in embryos depleted in their DFCs (DFC specific loss-of-function) in Vgll4l, Vgll4b, Yap, Taz, Tead1a and Tead3a morphants.

DOI: https://doi.org/10.7554/eLife.45241.004

**Figure supplement 2—source data 1.** Numerical data for *Figure 1—figure supplement 2*.

*Figure 1 continued on next page*

*Figure 1 continued*

DOI: https://doi.org/10.7554/eLife.45241.005
**Figure supplement 3.** Phenotype of *vgll4l*, *vgll4b*, *yap,taz* and *tead3a* homozygous mutant embryos.
DOI: https://doi.org/10.7554/eLife.45241.006

(*Figure 1B*) by incubating embryos in 2.5 µM of Verteporfin, a drug inducing Yap sequestration in the cytoplasm and promoting its degradation in the proteasome (*Wang et al., 2016*).

To validate our observations obtained with MO knockdowns or drug treatment, we generated CRISPR-Cas9 mutants for *vgll4l*, *vgll4b*, *yap* and *taz*. For each gene, we identified insertions/deletions (INDELs) leading to a premature stop resulting in truncated proteins lacking essential functional domains (*Figure 1C*). For *tead3a,* a mutant lacking an essential splice site leading to a premature end of translation was obtained from the Zebrafish Mutation Project (ZMP) (*Kettleborough et al., 2013*). Whereas individual homozygous mutants embryos for Tead3a, Yap, Taz, Vgll4b and Vgll4l display normal morphology at late developmental stages (*Figure 1—figure supplement 3*), analysis of their laterality fully confirmed the LR asymmetry defects observed in knockdown experiments (*Figure 1D*, *Figure 1—figure supplement 3*).

In the mouse embryo, partial redundancy of Yap and Taz has been proposed to explain the weak phenotype observed in Taz mutants (*Miesfeld et al., 2015*; *Sun et al., 2017*). In fish, whereas we observe mild laterality defects in *taz* morphants and mutants, these defects are much more severe in double Yap/Taz MO knockdowns (*Figure 1B*). In addition, lack of one copy of *yap* strongly increases the laterality defects observed in homozygous *taz* mutants (from 20% embryos with no heart jog or a right heart jog in single *taz* mutants to 55% for embryos homozygous for *taz* and heterozygous for *yap* - *Figure 1D*).

Altogether, the analysis of loss of function of Hippo TFs/TcoFs using a variety of approaches allowed us to uncover a novel and essential role for Hippo TFs/TcoFs in establishing LR asymmetry.

## Hippo TFs/TcoFs are required for the formation of the LRO

We then investigated whether the laterality phenotypes observed for the loss of function of these TFs/TcoFs resulted from defects in the formation and/or function of the LRO. Because Yap and Taz are partially redundant, we analyzed their function conjointly, using double Yap/Taz loss of function. First, using in situ hybridization for *sox17*, an early DFC marker, we found that DFC clusters are present at early gastrula stage for every Hippo TF/TcoF mutant/morphant tested (not shown), strongly supporting that these factors are not required for the specification of LRO progenitors. Conversely, lack of their activity has a dramatic effect on the formation of the KV (*Figure 2A–D*, *Figure 2—figure supplement 1*) with a strong decrease of its size (*Figure 2E*). In contrast, gain of Yap function through injection of its in vitro synthesized mRNA in a DFC specific manner (*Esguerra et al., 2007*; *Matsui et al., 2011*) results in an increase of the KV size associated with an increase in the number of KV cells (*Figure 2—figure supplement 2*).

During gastrulation, inWT embryos, DFCs proliferate, their number increasing from ~20 at the onset of gastrulation to ~50 in the differentiated KV. While the initial number of DFCs at early gastrula (60% epiboly) in embryos lacking Hippo TFs/TcoFs function is similar to control, we found a significant reduction in the DFC number at the end of gastrulation for each loss of function analyzed (*Figure 2F*). We then investigated whether the decrease in DFC number resulted from defects in cell proliferation and/or in cell survival. Measurement of the mitotic index at late gastrula stage revealed a moderate effect in Vgll4l, Yap/Taz and Tead1a knockdowns, while the mitotic index of the DFCs was not significantly affected in Vgll4b and Tead3a (*Figure 2G*). We also observed that cell survival, analyzed by measuring the apoptotic index of DFCs at 80% epiboly, was affected in Hippo TFs/TcoFs loss-of-function embryos (*Figure 2H*).

Finally, because motile cilia are essential to the function of the LRO, we looked for their presence and measured their length in embryos lacking activity of any of Hippo TFs/TcoFs. In all cases, we found a significant shortening of cilia (*Figure 3*) supporting that Hippo TFs/TcoFs are required for motile ciliogenesis.

There are contradictory reports on the role of Hippo signaling on ciliogenesis. In one study, Yap was described not to be involved in that process (*Kim et al., 2014*), whereas another study linked

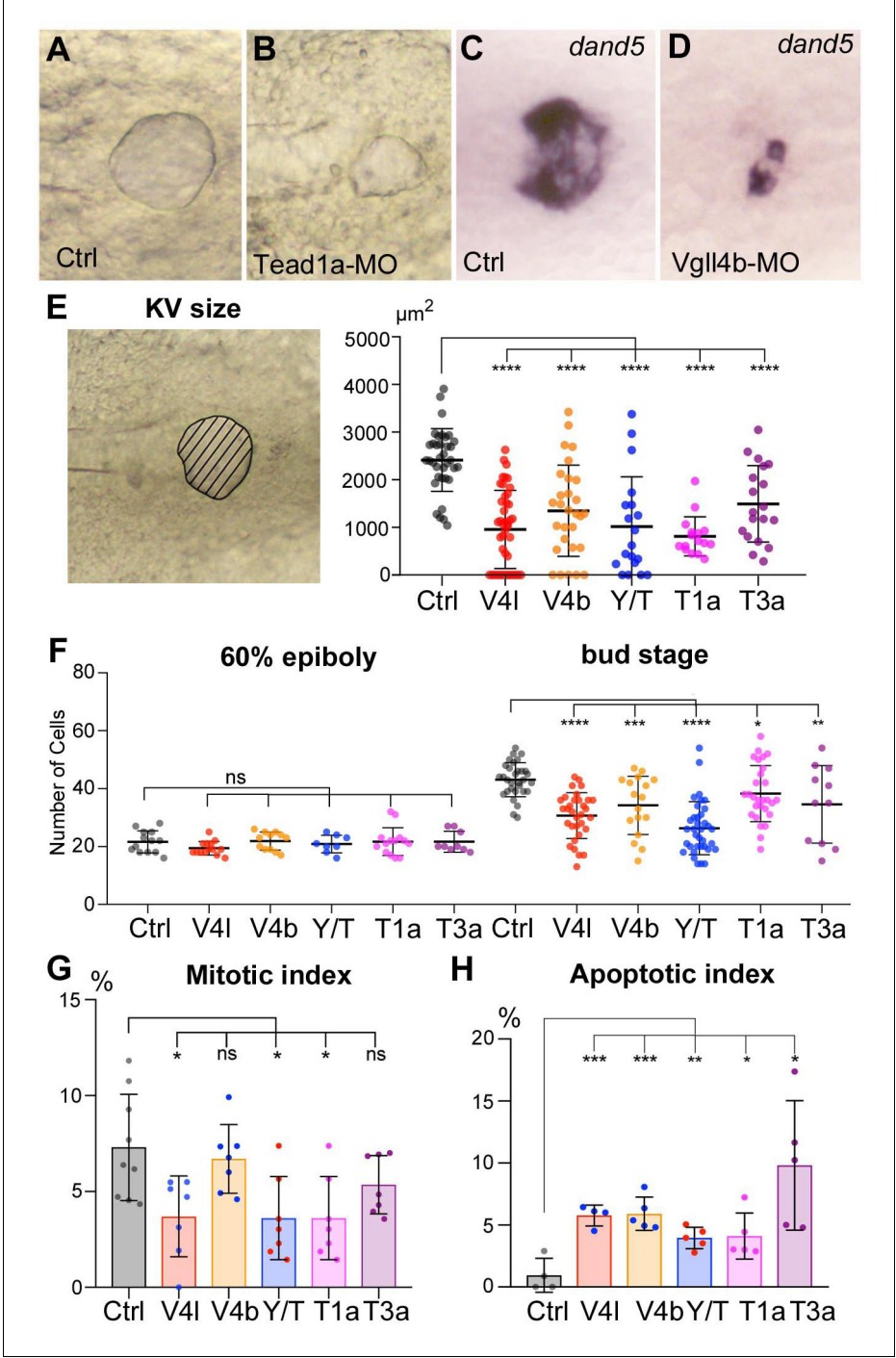

**Figure 2.** Loss of function of Hippo TFs/TcoFs affects the formation of the Left-Right Organizer. (A–D) Illustration of the strong decrease in the size of the KV at the 12-somite stage in loss-of-function conditions shown in brightfield for (A) a control embryo (Ctrl) and for (B) a TEAD1a morphant embryo (Tead1a-MO) and by in situ hybridization using a *dand5* probe in (C) Ctrl and in (D) Vgll4b morphant embryo (Vgll4b-MO). (E–H) Effect of Vgll4l (V4l), Vgll4b (V4b), Tead1a (T1a), Tead3a (T3a) loss of function and of Yap/Taz (Y/T) double loss of function on: (E) the size of the KV (expressed as the area of the planar projection of its lumen), (F) the number of DFCs present at early gastrula stage (60% epiboly) and at the end of gastrulation (bud stage), (G) the proliferation of the DFCs measured as their mitotic index at 75% of epiboly, (H) the survival of DFCs measured as their apoptotic index at 90% epiboly. In all cases control (Ctrl) embryos were injected with 8 ng of Standard MO. Graph indicates the mean of each experiment, error bars indicate standard deviation and dots indicate the individual

*Figure 2 continued on next page*

*Figure 2 continued*

measurement for DFC groups or individual KV in control and loss of function conditions. Statistical significance between controls and the different loss-of-function conditions: two-tailed unpaired t-test. *p≤0.05, **p≤0.01, ***p≤0.001, ****p≤0.0001. ns: not significant. Numerical data for (**E–H**) and details of statistical analysis are provided in *Figure 2—source data 1*.

DOI: https://doi.org/10.7554/eLife.45241.008

The following source data and figure supplements are available for figure 2:

**Source data 1.** Numerical data for *Figure 2E, F, G and H*.

DOI: https://doi.org/10.7554/eLife.45241.012

**Figure supplement 1.** KV defects in homozygous *vgll4l*, *vgll4b*, *yap*, *taz* and *tead3a* mutants.

DOI: https://doi.org/10.7554/eLife.45241.009

**Figure supplement 2.** Gain of function of Yap results in the formation of a larger KV.

DOI: https://doi.org/10.7554/eLife.45241.010

**Figure supplement 2—source data 1.** Numerical data for *Figure 2—figure supplement 2G and H*.

DOI: https://doi.org/10.7554/eLife.45241.011

Yap to the formation of non-motile cilia during zebrafish kidney development (*He et al., 2015*). Our observations support the latter conclusion, implicating Yap in ciliogenesis and demonstrating that the canonical Hippo pathway is required for proper organization of motile cilia.

## The TcoFs Yap/Taz and Vgll4l control the transcriptome of LRO progenitors

Because Vgll4l and Yap/Taz are TcoFs, they are expected to function by regulating gene expression. In a first experiment, we examined the consequences of their loss of function by MO knockdown on the expression of the 78 DFC specific genes we previously identified in high throughput in situ hybridization screens (http://zfin.org/, gene expression section). Expression of 30% of tested candidates (24/78) was either strongly decreased or completely absent in Vgll4l depleted DFCs (*Figure 4*).

Remarkably, 1/3 (8/24) of these genes are known to be required for the formation of the KV. They are involved in cilium assembly (*cdc14aa, daw1, dnaaf4, ttc25*), KV lumen expansion (*cftr, cldn5a*), proton transport (*atp6ap1b*) that has been shown to mediate DFC proliferation (*Gokey et al., 2015*), DNA methylation (*dnmt3bb.1*) or belong to the Nodal signaling pathway (*ndr1*). Similarly, we found that DFCs depleted for Yap/Taz display a clear decrease in expression for *cdc14aa, tnfrsf21, dnaaf4, cftr* or *ndr1* (not shown). Therefore, in addition to their known function in regulating cell proliferation and survival, Vgll4l and Yap/Taz regulate, directly or indirectly, the expression of genes that are essential for a variety of processes involved in the formation and function of the KV.

We investigated the tissue specificity of Vgll4l and Yap activity by performing DFC specific MO knockdowns for a subset of these 24 probes (*Figure 4—figure supplement 1*). In all cases we observed a strong decrease of the expression of these genes in the progenitors of the LRO. This shows that the regulation of the expression of DFC specific genes by Yap and Vgll4l we observed is not a secondary consequence of a global loss of function affecting the whole embryo from the beginning of development but results from the activity of these TcoFs in the DFCs.

We extended this study by establishing the transcriptome of LRO progenitors at late gastrula stage in Vgll4l or in Yap/Taz morphant embryos from the line *Tg(sox17:GFP)$^{S870}$* (*Sakaguchi et al., 2006*), which fluorescently labels DFCs. For each condition (control, Vgll4l or Yap/Taz loss of function), four DFC clusters were isolated and their RNA sequenced. Principal Component Analysis (PCA) was performed on transcriptome data of the four replicates for each condition (*Figure 5A*) and reveals that experimental replicates were highly reproducible and strongly clustered. Each experimental condition segregated in distinct groups pointing out that DFCs of control embryos and of embryos lacking Vgll4l or Yap/Taz have clearly distinct transcriptomes.

We found 9215 differentially expressed genes (DEGs: genes with normalized counts ≥ 1; | log2FoldChange| ≥ 1; Benjamini-Hochberg - False Discovery Rate - adjusted p-value<0.05) in DFCs of embryos depleted in Vgll4l and 7,925 DEGs in DFCs of embryos lacking both Yap and Taz activities (*Supplementary file 1*). As a quality control of RNA sequencing, we confirmed that the

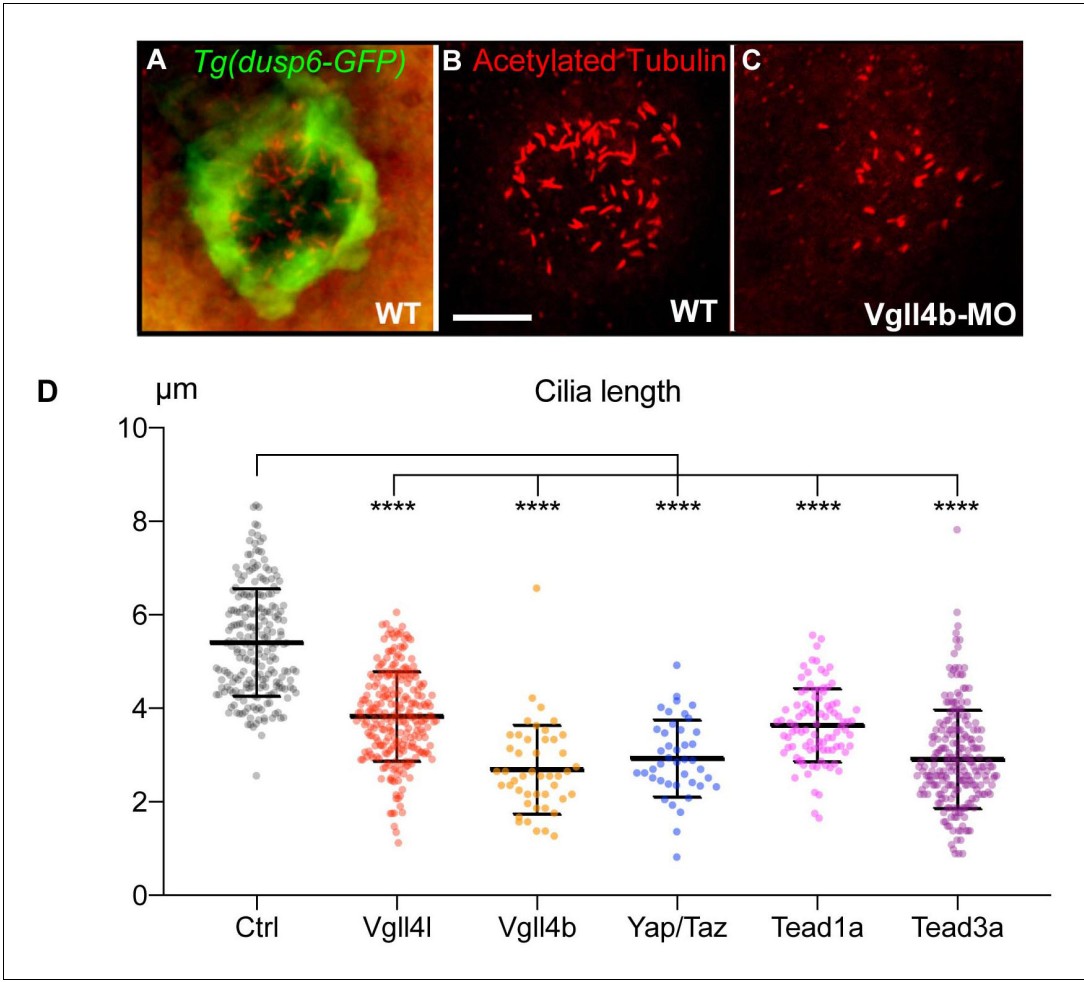

**Figure 3.** Loss of function of Hippo TFs/TcoFs leads to a reduction of the length of motile cilia of the KV. (**A–C**) Visualization of KV cilia at the 10-somite stage using an anti-acetylated tubulin antibody in (**A**) control embryos of the WT *Tg(dusp6:GFP)* line showing both KV cells (green) and cilia (red) or (**B**) only cilia. (**C**) cilia in Vgll4b morphant. Scale bar: 40 μm. (**D**) Length of KV cilia in Control (Ctrl) embryos and in Vgll4l, Vgll4b, Yap/Taz, Tead1a and Tead3a morphant embryos. Control (Ctrl) embryos were injected with 8 ng of Standard MO. Graph indicates the mean of cilia length, error bars the standard deviation. Statistical significance between controls and different loss-of-function conditions: two-tailed unpaired t-test. ****p≤0.0001. Numerical data and details of statistical analysis are provided in *Figure 3—source data 1*.

DOI: https://doi.org/10.7554/eLife.45241.013

The following source data is available for figure 3:

**Source data 1.** Numerical data for *Figure 3D*.

DOI: https://doi.org/10.7554/eLife.45241.014

expression of all the 24 genes found strongly decreased in Vgll4l loss of function in our in situ hybridization analysis (*Figure 4*) was also significantly downregulated in the transcriptome of Vgll4l depleted DFCs (*Figure 4—figure supplement 2*).

Analysis of the overlapping sets of DEGs for Vgll4l and Yap/Taz loss of function (6423 genes - *Figure 5B*) revealed that 84% (5394/6423) of them are regulated similarly in DFCs (either downregulated or upregulated) by Vgll4l and by Yap/Taz while only 16% (1029/6423) are regulated in opposing ways (upregulated in loss of function of Vgll4l while downregulated in Yap/Taz or vice-versa). This is a surprising observation as Vgll4 in other model systems is thought to act antagonistically to Yap/Taz by competition for the binding to TEADs (*Johnson and Halder, 2014*). Because the DEGs identified in the transcriptome analysis include both direct and indirect target genes, we examined the effect of Vgll4l and of Yap/Taz loss of function on the expression of zebrafish homologues of

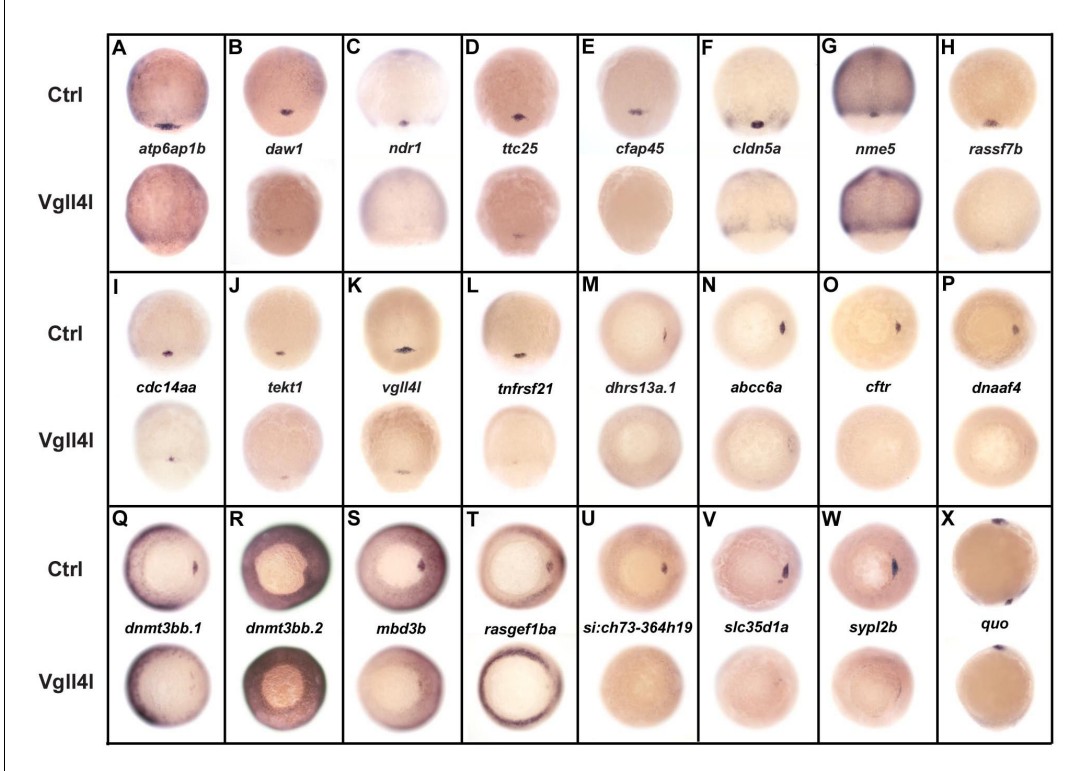

**Figure 4.** DFC specific genes downregulated in Vgll4l loss of function. Whole-mount in situ hybridization for genes that are expressed in DFCs at 70–90% epiboly in control (Ctrl) embryos and that are strongly downregulated in Vgll4l MO knockdown. Embryos are in dorsal view, animal pole to the top (**A–L**), in vegetal pole view dorsal to the right (**M–W**) and in lateral view anterior to the top dorsal to the right (**X**). Name of the genes probed is indicated in between control (top) and Vgll4l loss of function embryos (bottom).

DOI: https://doi.org/10.7554/eLife.45241.015

The following figure supplements are available for figure 4:

**Figure supplement 1.** Effect of DFC specific MO knockdowns on the expression of a selection of DFC specific genes.
DOI: https://doi.org/10.7554/eLife.45241.016
**Figure supplement 2.** Differential expression at late gastrulation of the genes analyzed by in situ hybridization in Vgll4l depleted embryos.
DOI: https://doi.org/10.7554/eLife.45241.017

direct target genes of Yap in mammals. We found 318 homologues for 380 Yap direct target genes described in mammals (*Lin et al., 2015*; *Wang et al., 2018*; *Zanconato et al., 2015*) present within the zebrafish genome (GRCz10). Amongst them 143 were DEGs for both Vgll4l and Yap/Taz. Analysis of the transcriptome data revealed (*Supplementary file 2*) that 82.5% of these genes (118/143) are regulated similarly by Vgll4l and Yap/Taz (61 downregulated and 57 upregulated in morphant DFCs) while only 17.5% (25/143) are regulated in opposite ways. It is very likely that most of the genes that are direct targets of Yap in mammals will also be direct targets of Yap in zebrafish. Therefore, the similar regulation of these genes by Vgll4l and by Yap/Taz is a strong evidence against a role of Vgll4l as a Yap antagonist during formation of the LRO in zebrafish.

To identify the biological function of DEGs for Vgll4l and Yap/Taz, we determined the functional categories of each gene by querying the Gene Ontology (GO) database (*Ashburner et al., 2000*; *The Gene Ontology Consortium, 2017*). Functional grouping of the GO terms based on GO hierarchy revealed that amongst the most prominent groups are those associated with the formation and function of the LRO: determination of LR symmetry, cilium movement, cilium organization and epithelium development (*Figure 5C*). In addition, mitotic cell cycle process was a GO term found for Yap/Taz while Vgll4l was associated with the GO term covalent chromatin modification, uncovering a role for this TcoF in the regulation of the expression of epigenetic factors.

Amongst the 166 genes whose loss of function has been described resulting into phenotypic defects in DFCs or in KV (http://zfin.org/, phenotype section), we found 134 genes expressed in

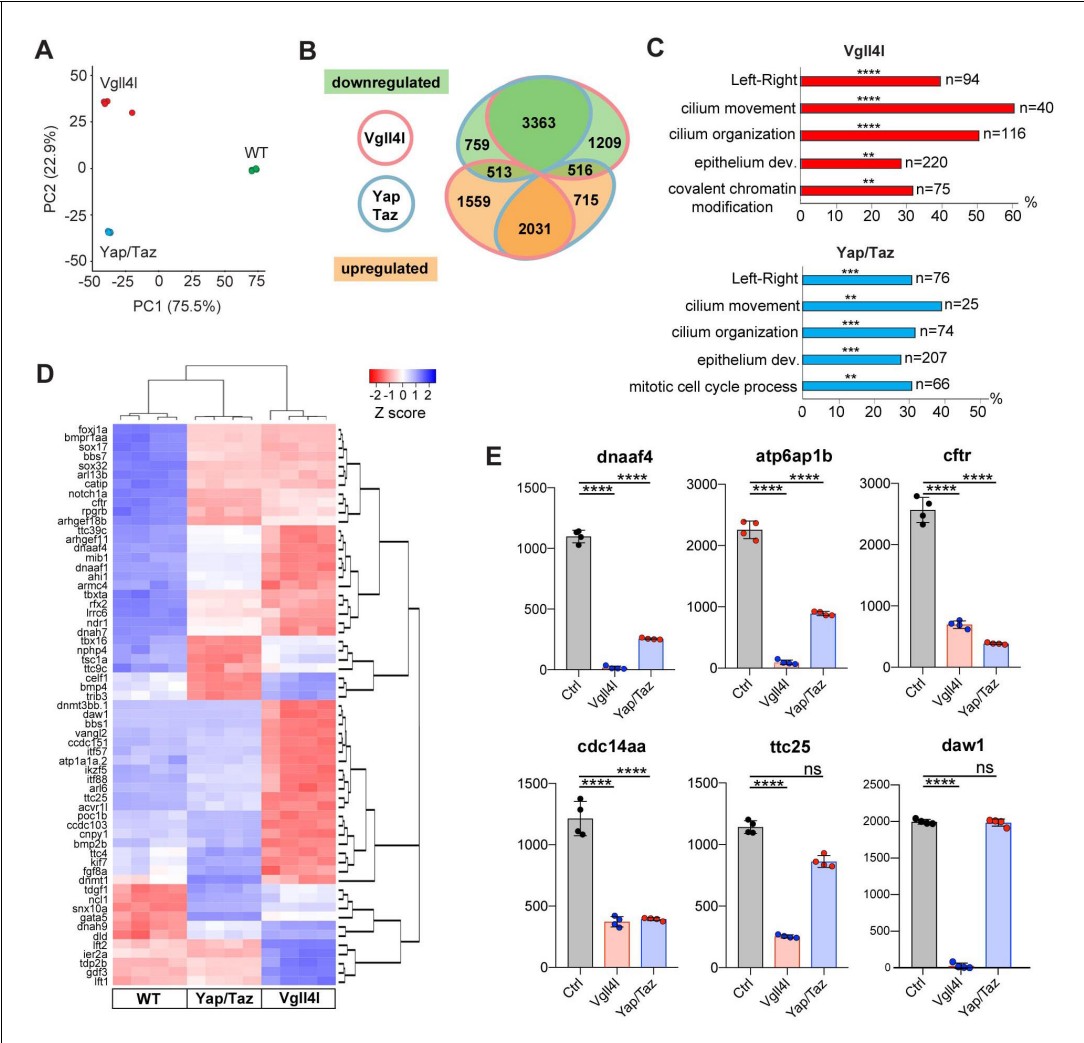

**Figure 5.** Transcriptome analysis of DFCs lacking Vgll4l or Yap/Taz function. (A) Principal component analysis (PCA) of DFC transcriptomes for Vgll4l morphants (red), Yap/Taz double morphants (blue) and control (green) showing that the four experimental replicates were highly reproducible and strongly clustered while each experimental condition segregated in distinct groups. (B) Venn diagram illustrating overlaps of differentially expressed genes (DEGs) between groups of DFCs lacking activity of Vgll4l (red line) or Yap/Taz (blue line). Downregulated genes (green), upregulated genes (orange). (C) Significant DEGs for Vgll4l (top, red) and for Yap/Taz (bottom, blue) were analyzed for selected biological processes. Bars represent the percentage of associated genes assigned to an unique GO term with the absolute number of associated genes located at the end of the bars (p-value corrected with Benjamini-Hochberg : **p≤0.01, ***p≤0.001, ****p≤0.0001). (D) Heatmap of DEGs whose loss of function is associated with DFCs and/or KV phenotypes. (E) Examples of genes downregulated in Vgll4l or in Yap/Taz depleted embryos: *dnaaf4* which is required for cilia movement (*Tarkar et al., 2013*), *atp6ap1b* known to mediate DFC proliferation (*Gokey et al., 2015*); *cftr* that controls KV lumen expansion (*Navis et al., 2013*); *cdc14aa* that contributes to ciliogenesis (*Clément et al., 2011*). *ttc25* that is critical for cilia formation and function (*Xu et al., 2015*) is downregulated in both Vgll4l and Yap/Taz loss of function but is not a DEG for Yap/Taz. Finally *daw1* that is essential for dynein assembly and ciliary motility (*Gao et al., 2010*) is downregulated only in Vgll4l loss of function. Graph bars indicate the mean expression expressed in normalized counts, the error bars the standard deviation and the dots, the value of each biological replicate. Statistical significance between controls and the different loss of function conditions: two-tailed unpaired t-test. ****p≤0.0001. ns: not significant (either p>0.05 and/or |log2FoldChange| < 1). Numerical data for (E) are available in *Supplementary file 1*.

DOI: https://doi.org/10.7554/eLife.45241.018

DFCs at late gastrula stage. Remarkably, 62% (83/134) and 50% (67/134) of these genes are respectively downregulated in loss of function of Vgll4l and Yap/Taz (*Figure 5D*, *Supplementary file 3*). This includes 75% (46/61) of genes involved in motile ciliogenesis and associated with defects of the KV function for Vgll4l depleted DFCs and 52% (32/61) for Yap/Taz loss of function (*Supplementary file 4*). Phenotypic analysis already revealed that these TcoFs are essential to

proper formation of KV motile cilia (*Figure 3*). Altogether, our transcriptome analysis further reinforced the conclusion that these factors play a major role in the regulation of motile ciliogenesis during the formation of the LRO.

## Yap/Taz and Vgll4l regulate the activity of signaling pathways involved in LRO formation

The molecular and cellular processes leading to the formation of a functional LRO are known to be regulated by major signaling pathways including Nodal, FGF, non-canonical Wnt and Notch (reviewed in *Matsui and Bessho, 2012*). We investigated the impact of Vgll4l and Yap/Taz on the expression of essential components (ligands, receptors, signal transductors, regulators) of these different pathways in DFCs at late gastrula stage. We found a strong downregulation of *nodal related 1* (*ndr1*) expression in both Vgll4l (*Figures 4C* and *6A*) and Yap/Taz (*Figure 6A*) loss of function. In Vgll4l, this loss of Ndr1 transcripts is associated with a strong decrease of the expression of *smad2a*, a R-Smad known to transduce Nodal signaling and with a strong upregulation of the expression of two Nodal feedback antagonists, *lefty1* (*lft1*) and *lefty2* (*lft2*) (*Agathon et al., 2001*). Both the decrease in Smad2a transcripts and the upregulation of Lft1 and Lft2 expression may disrupt the Nodal positive autoregulatory loop that is essential for the maintenance of *ndr1* transcription. In Yap/Taz loss of function, the strong decrease in *ndr1* expression is associated with a decrease in Smad3a transcripts, another R-Smad transducing Nodal signaling, but we did not observe an upregulation of Lft1 or Lft2 expression (*Figure 6A*).

In addition to Nodal, the non-canonical Wnt, the FGF and the Notch pathways are also affected. We observed a strong downregulation of expression of ligand (Wnt11) and receptors (Fzd8a, Fzd10 for Vgll4l, Fzd10 for Yap/Taz) of the non-canonical Wnt pathway (*Figure 6B*). Expression of ligands (Fgf8a and Fgf1a for Vgll4l; Fgf1a for Yap/Taz) of the FGF signaling pathway is downregulated in both Vgll4l and Yap/Taz loss-of-function conditions and the amount of transcripts of the Fgf receptor, Fgfr1a, and of the positive Fgf feedback regulator, Cnpy1, is strongly decreased in DFCs lacking Vgll4l (*Figure 6C*). As well the expression of ligands (Dla, Jag2b) and receptors (Notch1a, Notch1b and Notch3) of the Notch signaling pathway (*Figure 6D*) is downregulated in DFCs lacking either Vgll4l or Yap/Taz. Altogether, these observations provide evidence that Vgll4l and Yap/Taz are essential upstream regulators of the major signaling pathways controlling the formation of the LRO.

## Yap/Taz and Vgll4l control expression of TFs known to be required for LRO formation

The precise temporal order by which TFs selectively activate gene expression during development is critical to ensure proper lineage commitment, cell fate determination, and ultimately organogenesis. So far, only a small number of TFs has been shown to be necessary for the formation of the LRO, either for the specification of its progenitors [Dharma (*Fekany et al., 1999*), Sox32 (*Essner et al., 2005*), Sox17 (*Aamar and Dawid, 2010*)] or for the formation and function of the KV [FoxJ1a (*Yu et al., 2008*), Tbx16 (*Amack et al., 2007*), Tbxta (*Amack et al., 2007*) and Rfx2 (*Bisgrove et al., 2012*)]. We investigated the impact of loss of function of Vgll4l or Yap/Taz on the expression in LRO progenitors of these TFs. Because *dharma* is not expressed in DFCs at gastrula stage it is therefore not regulated in these cells by Vgll4l or Yap/Taz. However, transcripts of *sox32*, *sox17*, *tbxta*, *tbx16*, *foxj1a* and *rfx2* are expressed in DFCs during gastrulation and we found they are all significantly downregulated in Vgll4l and Yap/Taz loss of function (*Figure 7*). These observations strongly support that Vgll4l and Yap/Taz act upstream of the TFs known to be essential to the formation of the LRO and are involved in the regulation of their expression.

## Vgll4l regulates expression of writers and readers of DNA methylation marks in LRO progenitors

Lineage commitment, cell fate determination and ultimately organogenesis are also regulated by epigenetic mechanisms. Transcriptome analysis revealed that depletion of Vgll4l affects expression of genes involved in covalent chromatin modification (*Figures 5C* and *8*). Indeed, in DFCs, Vgll4l loss of function leads to a strong downregulation of the expression of genes coding for writers of the DNA methylation marks: the de novo DNA methyltransferases Dnmt3ba, Dnmt3bb.1, Dnmt3bb.2 (*Chédin, 2011*) (*Figures 4Q–R* and *8*) and the maintenance DNA methyltransferase

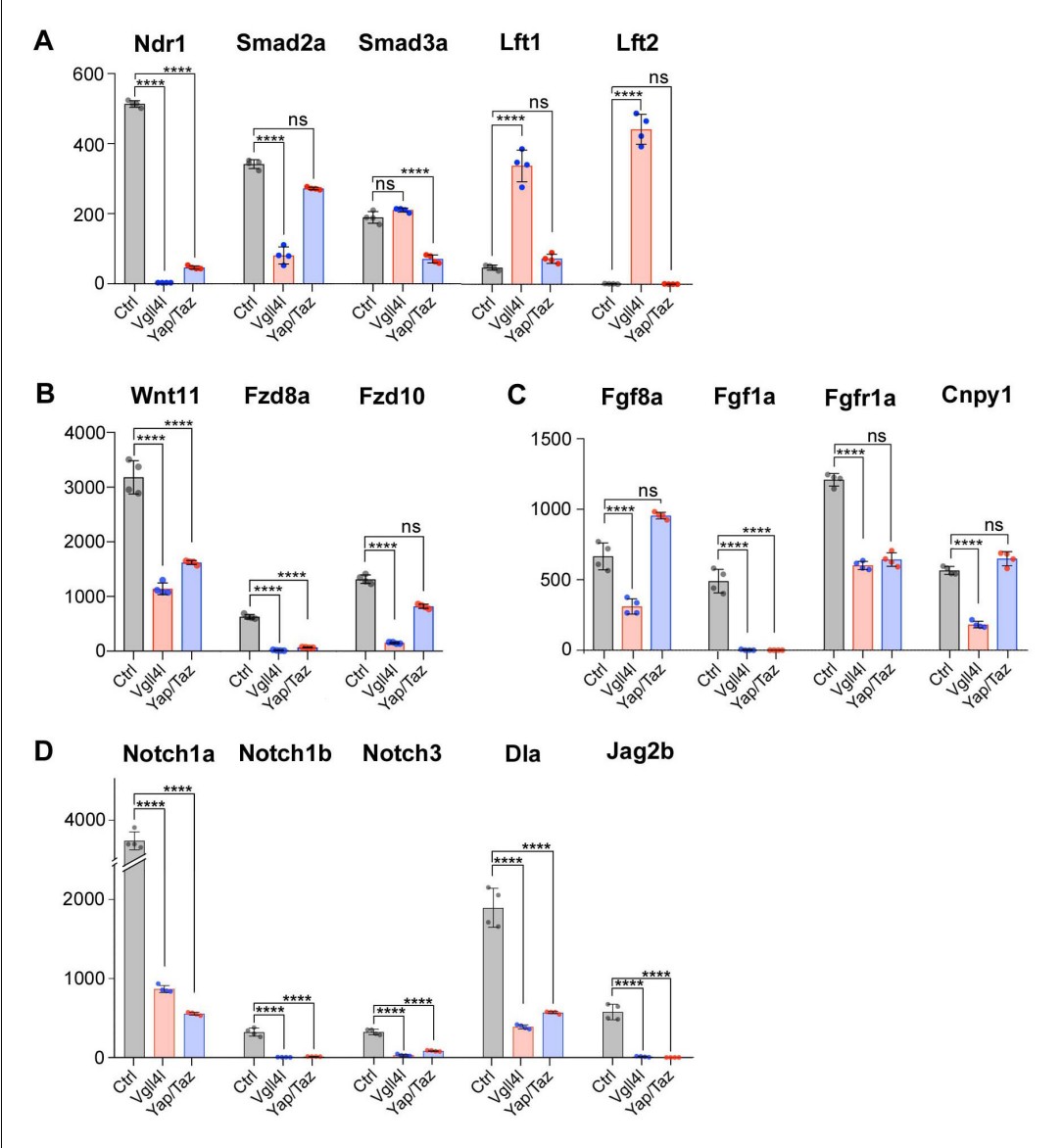

**Figure 6.** Regulation by Vgll4l or by Yap/Taz of critical genes from major signaling pathways controlling the formation of the KV. (A) Strong downregulation of *nodal related 1* (*ndr1*) expression in Vgll4l and Yap/Taz morphants. For Vgll4l, loss of Ndr1 transcripts is associated with a strong decrease of the expression of *smad2a*, a R-smad known to transduce Nodal signaling, as well as to a strong upregulation of the expression of Nodal feedback antagonists, *lefty1* (*lft1*) and *lefty2* (*lft2*). In Yap/Taz loss of function, the strong decrease in *ndr1* expression is associated with a decrease in expression of *smad3a*, another R-Smad transducing Nodal signaling, but not with an upregulation of *ltf1/2*. (B) There is a strong downregulation of expression of ligand (Wnt11) and receptors (Fzd8a and Fzd10 for Vgll4l, Fzd8a for Yap/Taz) of the non-canonical Wnt pathway. (C) Expression of ligands (Fgf8a and Fgf1a for Vgll4l; Fgf1a for Yap/Taz) and receptor (Fgfr1a) of the FGF signaling pathway is strongly downregulated in both Vgll4l and Yap/Taz loss-of-function conditions. Transcripts of the positive regulator (Cnpy1) are strongly decreased in Vgll4l loss of function. (D) The expression of ligands (Dla, Jag2b) and receptors (Notch1a, Notch1b and Notch3) of the Notch pathway is downregulated in DFCs lacking either Vgll4l or Yap/Taz. Bar graphs depict the mean expression of genes in DFCs of control (gray), Vgll4l (red) and Yap/Taz (blue) loss of function expressed as normalized counts with error bars indicating standard deviation. Dots indicate the individual value of each biological replicate (n = 4). Statistical significance between controls and the different loss-of-function conditions: two-tailed unpaired t-test. *p≤0.05, **p≤0.01, ***p≤0.001, ****p≤0.0001. ns: not significant (either p>0.05 or |log2FoldChange| < 1). Numerical data are available in *Supplementary file 1*.

DOI: https://doi.org/10.7554/eLife.45241.019

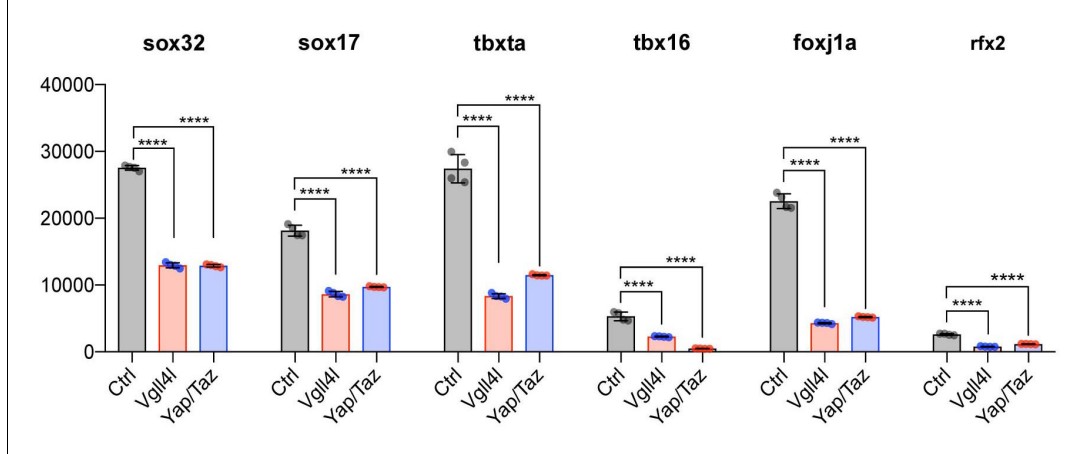

**Figure 7.** Vgll4l and Yap/Taz regulate expression of TFs required for KV formation. Transcripts of Sox32, Sox17, Tbxta, Tbx16, Foxj1a and Rfx2 are expressed in DFCs and are all significantly downregulated in Vgll4l and Yap/Taz morphants. The bar graphs depict the mean expression of genes in DFCs of control (gray), Vgll4l (red) and Yap/Taz (blue) loss of function expressed as normalized counts with error bars indicating standard deviation. Dots indicate the individual value of each biological replicate (n = 4). Statistical significance between controls and loss of function conditions: two-tailed unpaired t-test. *p≤0.05, **p≤0.01, ***p≤0.001, ****p≤0.0001. ns: not significant (p>0.05 and/or |log2 fold change| < 1). Numerical data are available in **Supplementary file 1**.

DOI: https://doi.org/10.7554/eLife.45241.020

Dnmt1 (*Mohan and Chaillet, 2013*) (*Figure 8*). In addition we found that Vgll4l also regulates the expression of the readers of DNA methylation marks, the Methyl-CpG-Binding Domain proteins: Mbd1b, Mbd2, Mbd3a (*Figure 8*), Mbd3b (*Figures 4S*, *8*) (note that while in mammals the methyl binding domain of MBD3 harbors critical mutations - K30H and Y34F - preventing it to bind to methylated DNA (*Saito and Ishikawa, 2002*), the methyl binding domain of Mbd3 in lower vertebrates retains the K30 and Y34 amino acids and binds to methylated DNA (*Bogdanović and Veenstra, 2011*) and Mbd6 (*Figure 8*). The other Dnmts and Mbds encoded by the zebrafish genome are either not expressed or expressed at low level in DFCs at late gastrula stage (*Supplementary file 1*).

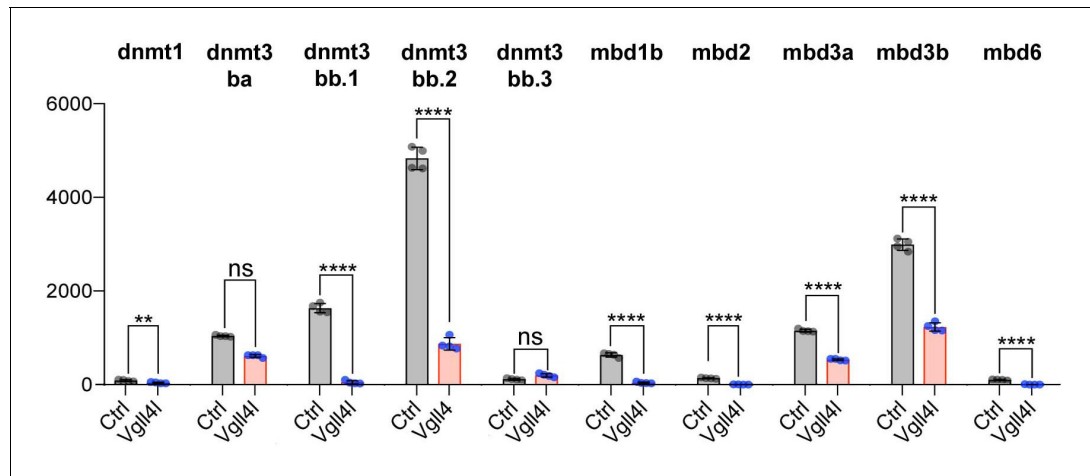

**Figure 8.** Vgll4l regulates expression of epigenetic factors writers and readers of DNA methylation marks. The bar graphs depict the mean expression of genes in DFCs of control (gray), Vgll4l (red) loss of function expressed as normalized counts with error bars indicating standard deviation. Dots indicate the individual value of each biological replicate (n = 4). Statistical significance between controls and loss of function conditions: two-tailed unpaired t-test. *p≤0.05, **p≤0.01, ***p≤0.001, ****p≤0.0001. ns: not significant (p>0.05 and/or |log2 fold change| < 1). Numerical data are available in **Supplementary file 1**.

DOI: https://doi.org/10.7554/eLife.45241.021

We investigated the possible role of these writers and readers of DNA methylation marks in the establishment of LR asymmetry by examining cardiac jogging defects in loss-of-function conditions (*Figure 9A*). Depletion of the de novo DNA methyltransferase, Dnmt3bb.1 has been previously shown to affect the establishment of the LR asymmetry (*Wang et al., 2017*). We confirmed this observation with a TALEN induced mutant that affects only the catalytic site of Dnmt3bb.1. Similarly, we found that loss of function of other de novo Dnmts expressed in DFCs (Dnmt3bb.2 and Dnmt3ba) or depletion of the readers of the DNA methylation marks, Mbd3a and Mbd3b also affect the establishment of the LR asymmetry (*Figure 9A*).

Phenotypic analyses of KV formation in embryos depleted for one de novo Dnmt (Dnmt3bb.1) or for one Methyl-CpG-Binding Domain protein (Mbd3b) reveals striking similarities with Vgll4l loss-of-function phenotype: reduction of KV size (*Figure 9B*), decreased number of DFCs at late gastrulation (*Figure 9C*) associated with a moderate decrease of DFC mitotic index (*Figure 9D*), an increased apoptotic index (*Figure 9E*) as well as a significant reduction in length of KV motile cilia (*Figure 9F*).

These observations strongly support that defects in KV organogenesis observed in Vgll4l loss of function may result, at least in part, from the decreased expression of Dnmt3bs and Mbd3s in DFCs. In strong agreement with this hypothesis, we found that laterality defects of Vgll4l loss of function can be partially rescued by gain of function of Dnmt3bb.1 (*Figure 9G*).

Because Vgll4l is necessary for Dnmt3bs expression we predict that its loss of function should impact DNA methylation of KV progenitors. To test this hypothesis we quantified DNA methylation in DFCs through an antibody-based detection of global nuclear DNA methylation using an anti 5-methylcytosine antibody (*Beaujean et al., 2018*). We found a significant decrease in DNA methylation of DFC nuclei in both Vgll4l morphants and in homozygous *vgll4l* mutants (*Figure 10*). This shows that Vgll4l regulates epigenetic programming in LRO progenitors by controlling in the DFCs the expression of epigenetic modifying enzymes.

## Discussion

Formation of organs during embryonic development requires both epigenetic modification that restricts lineage potential and the activation of tissue specific genes during the process of cell differentiation. Using the formation of the first functional organ of the zebrafish, the KV, as a model system we addressed the question of the transcriptional regulation of organ formation. Because this transient organ acting as the LRO in fish is very simple and is functional in just a few hours and from a well characterized population of progenitors, this model is particularly suitable to characterize how TFs and epigenetic modification control the differentiation of a ciliated epithelium during organogenesis. Since its description by Kupffer in 1867 (*Warga and Kane, 2018*) the KV has been studied in much details at the cellular and molecular levels. Genetic screens have identified 166 genes to be required for its formation and/or function (http://zfin.org/, phenotypes, Kupffer's vesicle). Surprisingly only 6 TFs: Rfx2, Foxj1a, Sox17, Sox32, Tbxta, Tbx16 have been reported to be necessary to the formation and function of the LRO. In the current study we identified six additional TFs and TcoFs required for this process. Our functional analyzes (morpholino knockdowns, drug treatment and Crispr-Cas9 mutants) reveal that the TcoFs known to mediate the transcriptional outcome of the Hippo signaling pathway (Yap and Taz), the DNA binding transcription factors they associate to (Tead1a and Tead3a) as well as Vgll4b and Vgll4l, two homologs of the mammalian Vgll4 that has been shown to negatively regulate Yap and Taz transcriptional activity, act as upstream regulators controlling the formation of the LRO of the zebrafish embryo. Each of these six genes is therefore essential for the establishment of the LR asymmetry of the embryo. Hippo TFs/TcoFs are not involved in the specification of LRO progenitors but are crucial for the regulation of their number at the end of gastrulation. However, while there is an effect on proliferation and viability of DFCs, most of the KV progenitors survived past gastrulation. The impact of Yap/Taz, Vgll4s and TEADs on the formation of the KV is not restricted to the final number of cells available for the formation of the vesicle but they also control the formation of the epithelium of the KV as well as cilia organization and motility.

Based on previous studies, mainly performed in pathological conditions, Vgll4 family members have been proposed to compete with Yap/Taz for TEADs, therefore inhibiting Yap/Taz function (*Jiao et al., 2014*; *Zhang et al., 2014*). However, during the process of differentiation of LRO progenitors, we found that Vgll4b and Vgll4l act similarly to Yap and Taz, regulating the number of

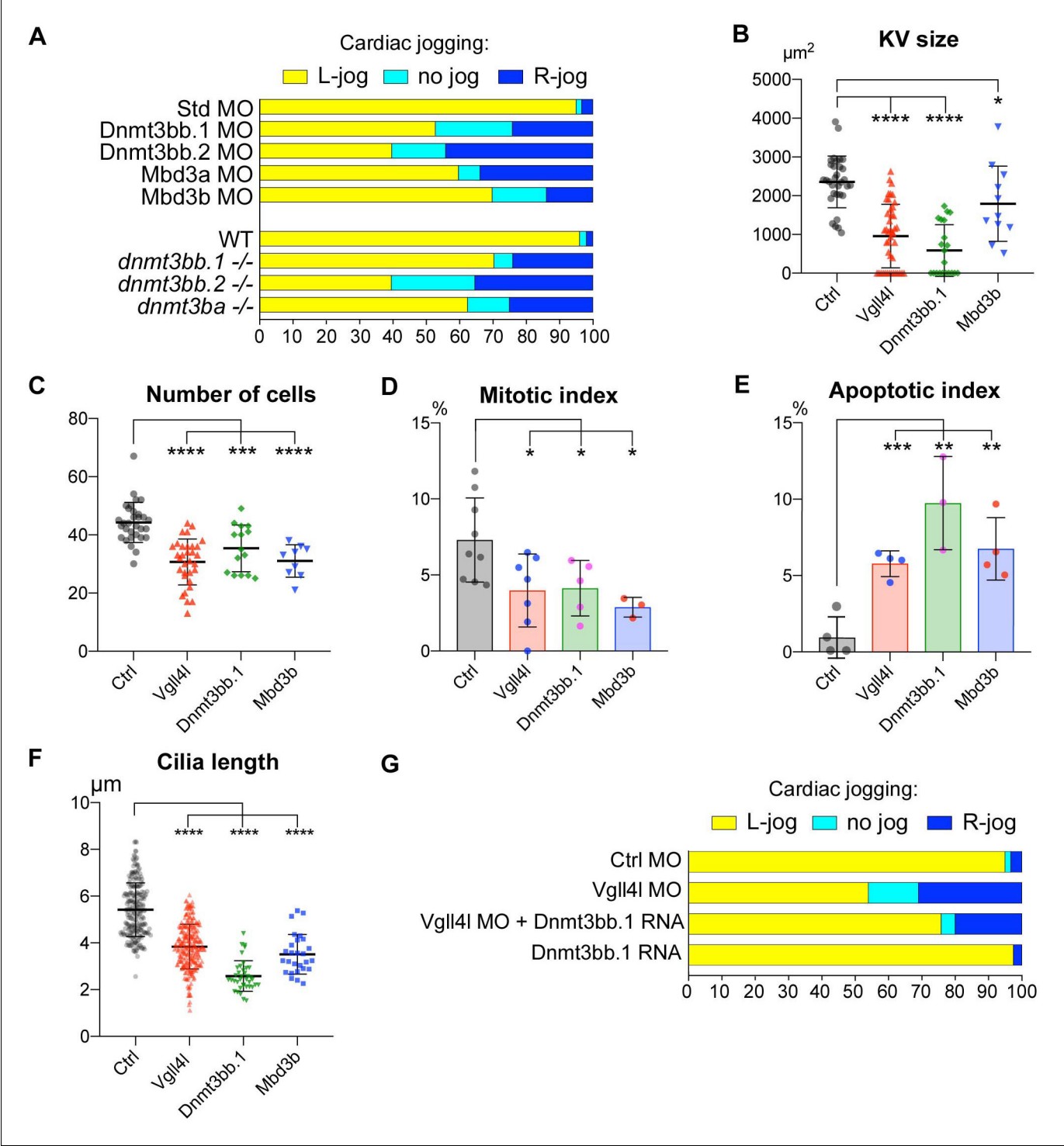

**Figure 9.** Similarity between phenotypes of Dnmt3bs, Mbd3s and Vgll4l loss of function in DFCs. (A) Loss of function of writers (*dnmt3s*) and readers (*mbds*) of DNA methylation marks strongly disrupt embryo laterality, analyzed by examining cardiac jogging at 25 hpf. Graphs indicate the percentage of embryos with normal Left jog (L jog - yellow), Right jog (R jog – dark blue) or no jog (light blue), at 25 hpf in control embryos (injected with 8 ng Standard - Std - MO), MO KDs (8 ng each MO) or in homozygous mutant embryos (-/-). (B–F) Phenotype analysis of LRO defects in morphants. (B) Size of the KV expressed in surface area of the planar projection of its lumen, (C) number of DFCs at the end of gastrulation, (D) Mitotic index at 80% epiboly, (E) Apoptotic index at gastrulation and (F) Length of KV cilia in : control (Ctrl) embryos (gray), Vgll4l (red), Dnmt3bb.1 (green) or Mbd3b (blue) MO knockdowns. Statistical significance between controls and loss of function conditions: two-tailed unpaired t-test. *p≤0.05, **p≤0.01, ***p≤0.001, *Figure 9 continued on next page*

Figure 9 continued

****p≤0.0001. Numerical data (**A–H**) and details of statistical analysis for (**B–F**) are provided in *Figure 9—source data 1*. (**G**) Injection of 100 ng of in vitro synthesized Dnmt3bb.1 RNA partially rescues laterality defects of Vgll4l morphants (8 ng Vgll4l MO), scored on cardiac jogging at 25 hpf.

DOI: https://doi.org/10.7554/eLife.45241.022
The following source data is available for figure 9:

**Source data 1.** Numerical data for *Figure 9A, B, C, D, E, F and G*.
DOI: https://doi.org/10.7554/eLife.45241.023

DFCs and controlling motile ciliogenesis. This strongly suggests that Vgll4s do not act as antagonists of Yap/Taz activity during formation of the LRO but that they rather function in the same or in parallel pathways.

Investigation of the impact of loss of function for Yap/Taz and for Vgll4l on the transcriptional network of the precursors of the LRO (both using a candidate gene approach and transcriptome analyzes) reveals that these transcription cofactors control expression of a large fraction (50% and 62% respectively) of the genes previously shown to be necessary for the formation and/or function of the LRO. This includes most of the genes required in the LRO for motile ciliogenesis (75% of them for Vgll4l, 52% for Yap/Taz). To the best of our knowledge, our study describes the first transcriptional networks during the formation of a ciliated epithelium. We also found that Hippo TFs/TcoFs regulate essential factors of major signaling pathways that have been shown to regulate KV precursor cell behaviors (clustering, collective migration, rosette formation, luminogenesis) during gastrulation and early somitogenesis until the formation of the fully functional LRO [reviewed in *Matsui and Bessho (2012)*].

Amongst the genes whose expression requires Yap/Taz and Vgll4l activities are essential factors (ligands, receptors, positive or negative regulators) of the major signaling pathways (Nodal, FGF, non-canonical Wnt and Notch/Delta) known to be critical to the formation of the LRO. This observation therefore places Yap/Taz and Vgll4l upstream of these signaling pathways and strongly supports that these TcoFs are master regulators of the formation and function of the LRO.

The activity of the Nodal signaling pathway has been shown to be modulated through DNA methylation of the regulatory sequences of the Nodal antagonist Lefty (*Dai et al., 2016*) (*Wang et al., 2017*). In WT embryos, *lefty* genes are not expressed in the precursors of the LRO. However, our transcriptome analysis revealed expression of *lft1* and *lft2* in DFCs of Vgll4l morphant embryos associated with a strong reduction of *ndr1* expression (*Figure 6*). In these Vgll4l depleted embryos, we also found strong downregulation of genes coding for the 'de novo' DNA methyltransferases that establish the initial DNA methylation pattern (*Figure 8*). Expression of Nodal is also strongly downregulated in embryos devoid of Yap/Taz activity. However, expression of *lft1* and *lft2* is not observed in the DFCs of Yap/Taz morphant embryos and genes coding for Dnmts were not downregulated in Yap/Taz loss of function. These observations show (1) that *ndr1* expression and activity are controlled in the DFCs by Vgll4l and Yap/Taz and (2) strongly suggest that distinct mechanisms are used by Vgll4l and Yap/Taz to regulate Nodal signaling in the DFCs, with Vgll4l repressing *lft1* and *lft2* expression likely through a positive control of the expression of de novo DNA methyltransferases in the progenitors of the LRO. The effect of Vgll4l loss of function on DNA methylation of DFC nuclei strongly support this interpretation.

The role of DNA methylation in chromatin modifications associated with cell differentiation has been extensively analyzed. However, the upstream regulation of this process is far less understood. Our study reveals that one transcription cofactor, Vgll4l, is required in the progenitors of the LRO for the expression of epigenetic factors, writers (the de novo DNA methyltransferase 3) and readers (Methyl-CpG binding domain proteins) of DNA methylation marks. This identifies another level of regulation of epigenesis during embryonic development through the tissue specific control of the expression of major epigenetic actors. The demonstration that Vgll4l regulates epigenetics in a temporal and spatial manner by affecting the expression of some of its major actors opens a new field of exploration for the understanding of the regulation of gene expression during cell differentiation.

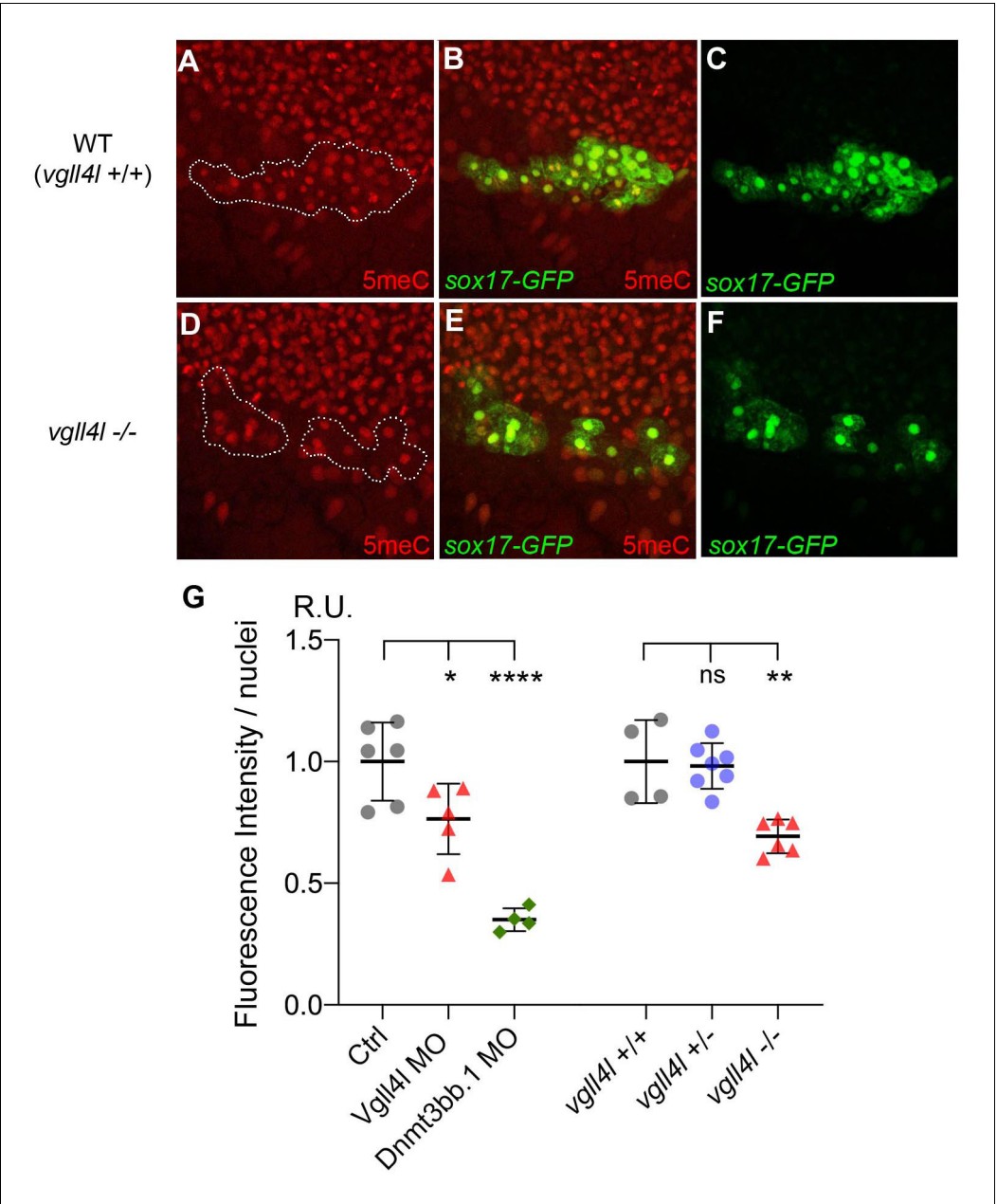

**Figure 10.** Effect of Vgll4l and Dnmt3bb.1 loss of function on DNA methylation in DFC nuclei. (A–F) Immunolabeling of DFC nuclei of embryos of the *Tg(Sox17:GFP)* strain with antibodies to five methyl Cytosine (5meC) in (A–C) WT and (D–F) *vgll4l* homozygous mutant. Dotted lines delimit the DFC clusters. (G) Quantification of DNA methylation measured by immunofluorescence intensity. The values on the graph correspond to the mean of 5meC fluorescence intensity per nuclei (FI/nuclei) quantified with the ImageJ software on all nuclei of DFC clusters in morphants for Vgll4l and Dnmt3bb.1 and in embryos from a cross between two heterozygous Vgll4l mutants that have been individually genotyped after measurement of FI/nuclei of their DFCs. Fluorescence intensity per nuclei is expressed in relative units (R.U.). Statistical significance between controls (standard MO or *vgll4l* +/+) and loss of function conditions: two-tailed unpaired t-test. ns, not significant (p>0.05), *p≤0.05, **p≤0.01, ***p≤0.001, ****p≤0.0001. Numerical data and details of statistical analysis for (G) are provided in *Figure 10—source data 1*.

DOI: https://doi.org/10.7554/eLife.45241.024

The following source data is available for figure 10:

**Source data 1.** Numerical data for *Figure 10G*.

DOI: https://doi.org/10.7554/eLife.45241.025

# Materials and methods

## Zebrafish line and husbandry

Zebrafish lines used in this study are the *AB/Tü* WT line, *Tg(sox17:GFP)$^{S870}$* (**Chung and Stainier, 2008**) and *Tg(dusp6:GFP)$^{pt19}$* (**Wang et al., 2011**). *vgll4l$^{va1}$* (an allele carrying a deletion of 7 nucleotides in the second exon resulting in a frame shift after amino-acid 36), *vgll4b$^{va2}$* (an allele carrying an INDEL - Δ4,+6 - resulting in a frame shift after amino-acid 109), *yap1$^{va3}$* (an allele of yap carrying a deletion of 8 nucleotides resulting in a frame shift after amino-acid 124), *wwtr1$^{va4}$* (an allele of taz carrying a deletion of 4 nucleotides resulting in a frame shift after amino-acid 85), *dnmt3bb.1$^{mk24}$* (an allele carrying a deletion of 6 nucleotides removing Ser 612 and Pro 613, two amino-acids of the catalytic site), *dnmt3bb.2$^{mk23}$* (an allele carrying an InDel - Δ5+13 – resulting in a frame shift after amino-acid 1237 and that lack the catalytic site) and dnmt3bamk26 (an allele carrying a 12 nucleotides deletion in the catalytic domain). Position and sequence of the different mutations are presented in **Supplementary files 6–9**.

This study was carried out in strict accordance with the recommendations in the Guide for the Care and Use of Laboratory Animals and all steps were taken to minimize animal discomfort. The University of Virginia Institutional Animal Care and Use Committee approved all protocols.

## In situ hybridization

In situ hybridization on whole-mount embryos have been performed as described in **Thisse and Thisse (2014)**; **Thisse and Thisse (2008)**. Conditions for the synthesis of antisense RNA probes for the following cDNA clones : *abcc6a* (**Li et al., 2010**) ; *atp6ap1b* (MGC:103523) ; *cdc14aa* (MGC:63654) ; *cldn5a* (MGC:85723) ; *daw1* (IMAGE:6904039) ; *dhrs13a.1* (cb464) ; *dnaaf4* (MGC:77853) ; *dnmt3bb.1* (cb633) ; *mbd3a* (IMAGE:7139207), *mbd3b* (cb99) ; *nme5* (MGC:92812) ; *quo* (cb9) ; *rasgef1ba* (MGC:66487) ; *rassf7b* (cb262) ; *si:ch73-364h19* (cb911) ; *slc35d1a* (IMAGE:7156746) ; *sox17* (MGC:91776) ; *sypl2b* (IMAGE:7136581) ; *tead1a* (MGC:63696) ; *tead3a* (CB882) ; *tekt1* (MGC :101797) ; *ttc25* (MGC:56362) ; *vgll4l* (cb747) and *yap1* (cb194) are available at http://zfin.org/ in the gene expression section.

The cDNAs of *ndr1* (**Agathon et al., 2003**) and *dand5* were inserted in the pBS-IISK+ vector, the cDNA of *cftr* (MGC:198381) was inserted in the pCR-XL-TOPO reverse vector as well as of *vgll4b* and *wwtr1* (*taz*) that were inserted in pCR2.1 TOPO vectors, PCR amplified using M13 forward and reverse primers and their antisense RNA synthesized using the T7 RNA polymerase. cDNA for dnmt3bb.2 inserted in pCR2.1 TOPO vector was PCR amplified using M13 forward and 5'- GGA TCCATTAACCCTCACTAAAGGGAAGACAGGAAACAGCTATGACC-3' primers and its RNA synthesized using the T3 RNA polymerase. cDNAs for *cfap45* (MGC: 158569), *tnfrsf21* (IMAGE:2602431) inserted in pME18S-FL3 vector were PCR amplified using 5'-TGTACGGAAGTGTTACTTCTGCTC-3' and 5'-GGATCCATTAACCCTCACTAAAGGGAAGGCCGCGACCTGCAGCTC-3' primers and the antisense RNAs synthesized using the T3 RNA polymerase.

Representative images were acquired using a coolsnap camera on a Leica macroscope.

## Morpholino knockdowns

Morpholinos (Gene Tools) were resuspended in sterile water as a 4 mM stock solution and diluted in 0.2% Phenol Red and 0.1M KCl before use to the appropriate concentration. Embryos were dechorionated at the one-cell stage using Pronase E and injected with 1 nl of morpholino solution, using an Eppendorf 5426 microinjector.

For knockdowns in the whole embryo, morpholino solutions were injected in the yolk in a position close to the blastomeres at the 1- to 4 cell stages. For DFC specific knockdown, injections were performed in the yolk close to the blastomeres at the 256- to 1 K-cell stage as described in **Wang et al. (2013)**. Names, sequences and amounts of MO injected in knockdowns and DFC specific knockdowns are provided in **Table 1**. In all experiments control embryos were injected with 8 ng of standard MO.

## Antisense oligonucleotides (ASO) knockdowns

Two non overlapping 2' O-Methyl RNA antisense oligonucleotides targeting the 5' end of Vgll4l coding sequence were synthesized by Integrated DNA Technologies.

**Table 1.** Name, sequence and amount of MO used in knockdown and in DFC targeted knockdown.

| MO name | Sequence | References | Amounts injected for knockdown | Amounts injected for DFC targeted knockdown |
|---|---|---|---|---|
| Vgll4l MO1 | TGTAGTGGAAATTAGTGACCGCCAT | This study | 8 ng | |
| Vgll4l MOsp | TTGGGCTGTCCTGTGAAAAGATGAG | This study | 6 ng | |
| Vgll4b MO1 | ACAGGTCCATTTTGGTAAAAAGCAT | (*Melvin et al., 2013*) | 4 ng | 8 ng |
| Vgll4b MO2 | AATCGCAGAAAGAGCAGCTTCTCTT | This study | 4 ng | |
| Yap MO1 | CTCTTCTTTCTATCCAACAGAAACC | (*Hu et al., 2013*; *Jiang et al., 2009*) | 6 ng | |
| Yap MOsp | AGCAACATTAACAACTCACTTTAGG | (*Skouloudaki et al., 2009*) | 4 ng | |
| Taz MO1 | CTGGAGAGGATTACCGCTCATGGTC | (*Hong et al., 2005*) | | 8 ng |
| Taz MOsp1 | TGTATGTGTTTCACACTCACCCAGG | This study | 6 ng | |
| Taz MOsp2 | ATGTGACTGCACAACAAACACAGAA | This study | 6 ng | |
| Tead1a MO1 | CATGGCAATGGATGTGATCTCAGAG | This study | 8 ng | |
| Tead1a MO2 | TGAGCCTGGAGAACTCAAGGCACAC | This study | 8 ng | |
| Tead3a MO1 | CGTCCATTCCGGTTTTGTCCATCCC | This study | | 2 ng |
| Tead3a MOsp1 | CAGCTTTCTGTTACTCACCATACAT | This study | 8 ng | |
| Tead3a MOsp2 | GGGTCTGAAATACTCACTCCTGAGA | This study | 8 ng | |
| Dnmt3bb.1 MO1 | TTATTTCTTCCTTCCTCATCCTGTC | (*Huang et al., 2013*; *Shimoda et al., 2005*) | 8 ng | 8 ng |
| Dnmt3bb.1 MOsp | CTCTCATCTGAAAGAATAGCAGAGT | (*Gore et al., 2016*) | 6 ng | 6 ng |
| Dnmt3bb.2 MO1 | CTCCGATCTTTACATCTGCCACCAT | (*Huang et al., 2013*; *Shimoda et al., 2005*) | 6 ng | 6 ng |
| Dnmt3bb.2 MOsp | GCACCTGAAAAAGTGTAAACACCAT | This study | 6 ng | 6 ng |
| Mbd3a MO | CCACCTTTTCCTCTCCATGATTTTC | (*Huang et al., 2013*) | 8 ng | 4 ng |
| Mbd3b MO | TCGTTTTTCTCCATCTCGCATTCTC | This study | 8 ng | 4 ng |
| Standard control MO | CCTCTTACCTCAGTTACAATTTATA | Gene tools | 8 ng | 8 ng |

DOI: https://doi.org/10.7554/eLife.45241.026

Vgll4l 2'OMe ASO-1: mG*mU*mG*mG*mA*A*A*T*T*A*G*T*G*A*C*mC*mG*mC*mC*mA

Vgll4l 2'OMe ASO-2 : mC*mU*mG*mC*mU*C*A*T*C*C*T*G*G*T*T*mA*mU*mG*mU*mA mA, mU,mC,mG: 2'O-Methyl(2'OMe)-modified RNA nucleotides; * phosphorothioate bond. ASO-1 and ASO-2 are respectively complementary to nucleotides 2–22 and 23–43 of the *vgll4l* open reading frame.

Knockdowns were performed as described in *Pauli et al. (2015)* by injection of 75 ng of each ASO into WT embryos at the 1- to 2 cell stage.

## Sense RNA synthesis and injection

PCR amplified fragments containing the complete open reading frame of Vgll4l, Vgll4b, Yap, Taz or Dnmt3bb.1 were cloned into the pCS2+ vector. For mRNA synthesis, constructs were linearized with NotI and transcribed using SP6 RNA polymerase using the mMESSAGE mMACHINE kit (Ambion). In vitro synthesized sense RNAs were injected either alone or in combination with MOs in rescue experiments. DFCs specific gain of function have been performed by injecting 0.4 µg of in vitro synthesized mRNA in the yolk of embryos at the 256–512 cell stages as described in *Esguerra et al. (2007)*; *Matsui et al. (2011)*.

## Crispr/Cas9 mutagenesis

Target sequences for CRISPR/Cas9 were identified using the optimized CRISPR Design – MIT (http://crispr.mit.edu/). Complementary primers for the target sites were annealed and ligated into BSA1-cleaved pDR274 plasmid (*Hwang et al., 2013*) (plasmid # 42250 from Keith Joung, obtained from Addgene). Plasmids for sgRNAs were linearized using Dra1 enzyme and sgRNAs synthesized with the Maxiscript-T7 kit from Ambion. As described in *Burger et al. (2016)* RNP complexes were formed by incubating 900 ng/µl of Cas9 protein (New England Biolabs) with 150 ng/µl of sgRNA in 300 mM KCl for 5 min at 37°C. The complexes were then injected into 1 cell stage embryos. Adult F0 fish were outcrossed to WT fish. Then, genomic DNA was extracted from fin clips of adult F1 individuals (*Meeker et al., 2007*). Targeted region was amplified by PCR and analyzed for INDELs using an heteroduplex mobility assay (*Ota et al., 2013*). Sequence of oligonucleotides used is provided in *Supplementary file 4*. Position and sequence of mutants is provided in *Supplementary files 6–9*.

## TALEN mutagenesis

TALEN sequences for mutagenesis of Dnmt3bb.1, Dnmt3bb.2 and Dnmt3ba were selected using Targeter 2.0 software (*Doyle et al., 2012*). TAL repeat assembly was achieved using the Golden Gate assembly method, and assembled repeats were integrated into the GoldyTALEN scaffold (*Bedell et al., 2012*; *Cermak et al., 2011*). Assembled vectors served as templates for in vitro mRNA transcription using the T3 mMessage mMachine kit (Ambion) according to manufacturer's instructions. 50–100 pg mRNA was injected into WT embryos at the one-cell stage. Position and sequences of mutants is provided in *Supplementary files 8* and *9*.

## Immunohistochemistry and imaging

Embryos were fixed overnight at 4°C in 4% paraformaldehyde then washed 3 times for 20 min in a PBS medium containing 1% Triton X100 (PBS-1%Triton) for 20 min. Embryos were incubated for 1 hr in the blocking buffer (PBS-1%Triton, sheep serum 10%) then overnight at 4°C in a medium containing the primary antibody: anti-Acetylated Tubulin (Sigma-Aldrich T7451), anti Phospho Histone H3 (Ser10) (Cell Signaling, 9701) or anti Cleaved Caspase 3 (Asp175) (Cell Signaling, 9661) used at 1:400 dilution, anti Sox17 (Novus, NBP2-24568), anti Yap (Cell Signaling, 4912) used at 1:200 dilution and anti 5-methyl-cytosine (abcam, ab10805) used at 1:10,000 dilution in the blocking buffer. Embryos were then washed 3 × 20 min with PBS-1%Triton and incubated 2 hr at room temperature in a medium containing the secondary antibody: a goat anti-rabbit Alexa Fluor 488 (Thermofisher, A11008) or a goat anti-mouse Alexa Fluor 546 (Thermofisher, A11030) used at a 1:800 dilution in the blocking buffer, and 2% Hoechst 33342 (Sigma-Aldrich) to label the nuclei. After three final washes of 20 min in PBS-1%Triton, embryos were mounted in 2% low melting agarose. Representative images were acquired using a Leica TCS LSI confocal macroscope. Images were analyzed using Image J software.

## Quantitative analysis of Kupffer's vesicle size, cilia length and number of DFCs

KV size, cilia length and DFCs were quantified as described in *Gokey et al. (2015)*; *Gokey et al. (2016)*. Embryos were observed in brightfield using a Leica macroscope and the area of the KV lumen was measured using the ImageJ software (NIH). For cilia, embryos immunostained with acetylated-tubulin antibodies were imaged using a Leica TCS LSI confocal macroscope and the length of cilia was measured using ImageJ software. The number of DFCs was determined using the *Tg(sox17: GFP)*$^{S870}$ by manually counting the number of Hoechst 33342 labeled nuclei of GFP+ DFCs in a Z-series of images collected using a Leica TCS LSI confocal macroscope.

For statistical analyses, P values were calculated with Graphpad prism eight software using two-tailed unpaired t-test. All raw data are available in *Figure 2—source data 1*, *Figure 3—source data 1* and *Figure 9—source data 1*.

## Pharmacological treatments

Verteporfin treatment: a 2 mg/ml stock solution of Verteporfin (SML0534, Sigma) was prepared in dimethylsulfoxide (DMSO). Embryos were incubated from the 1 cell stage to 24 hpf with verteporfin diluted in 0.3 x Danieau buffer at a final concentration of 2.5 µM.

## RNA extraction, cDNA library preparation and RNA-Seq

DFCs were dissected out in DMEM/F-12 medium (Gibco Dulbecco's Modified Eagle Medium: Nutrient Mixture F-12) from embryos of the Tg(sox17:GFP) line at the 90% epiboly stage injected at the 1 cell stage with 8 ng of standard MO (Control), 4 ng of Vgll4l MOsp or together with 4 ng of Yap MO1 and 4 ng of Taz MOsp1. Individual samples were transferred in a 0.5 ml Eppendorf tube with 2 µl of DMEM/F12. Total RNA was extracted and purified using SMART-Seq v4 Ultra Low Input RNA Kit for Sequencing from Clontech. Purified RNA samples were then reverse transcribed into cDNA and amplified. Libraries were constructed using a NEB DNA Ultra library construction kit (New England Biolabs), with standard TruSeq-type adapters. Libraries size and concentration were assessed using an Agilent 2100 Bioanalyzer. Libraries were multiplexed and 50 bp single-end were sequenced (Beijing Genomics Institute) on illumina Hiseq 2000 sequencer generating a minimum of 20 million reads per sample. Genome-wide transcriptome were produced from quadruplicate biological replicates.

## Bioinformatics

Bioinformatic analysis was performed by UVA bioinformatics core facility. Sequences alignment was done using STAR. Reads were mapped to GRCz10 Ensembl genes using the featureCounts software. DESeq2 Bioconductor package was used to normalize count data, estimate dispersion, and fit a negative binomial model for each gene. The Benjamini-Hochberg False Discovery Rate procedure was used to re-estimate the adjusted p-values for Ensembl gene IDs mapping to known genes. GO-term analysis was done in cytoscape 3.2.2 using the cluego plugin (*Bindea et al., 2009*). Heatmap was generated using heatmap.2 plugin in R.

## Acknowledgements

We thank HJ Yost for the *Tg(sox17-GFP)*[S870] and JD Amack for the *Tg(dusp6:GFP)*[pt19] lines, SD Turner and AT Nguyen for advices.

## Additional information

### Funding

| Funder | Grant reference number | Author |
|---|---|---|
| Centre National de la Recherche Scientifique | | Christine Thisse<br>Bernard Thisse |
| Institut National de la Santé et de la Recherche Médicale | | Christine Thisse<br>Bernard Thisse |
| University of Virginia | | Christine Thisse<br>Bernard Thisse |
| National Science Foundation | 1455901 | Bernard Thisse |
| National Institutes of Health | R01GM132131 | Christine Thisse<br>Jonathan Fillatre |
| National Institutes of Health | R01GM110092 | Mary Goll |

The funders had no role in study design, data collection and interpretation, or the decision to submit the work for publication.

### Author contributions

Jonathan Fillatre, Formal analysis, Validation, Investigation, Methodology; Jean-Daniel Fauny, Validation, Investigation, Visualization, Methodology; Jasmine Alexandra Fels, Investigation; Cheng Li, Mary Goll, Resources, Generated and characterized Dnmt3bs mutants; Christine Thisse, Conceptualization, Data curation, Supervision, Validation, Writing—original draft, Writing—review and editing; Bernard Thisse, Conceptualization, Data curation, Formal analysis, Supervision, Funding acquisition, Validation, Writing—original draft, Writing—review and editing

## Author ORCIDs

Mary Goll  http://orcid.org/0000-0001-5003-6958
Bernard Thisse  https://orcid.org/0000-0002-8365-1081

## Ethics

Animal experimentation: This study was performed in strict accordance with the recommendations in the Guide for the Care and Use of Laboratory Animals of the National Institutes of Health. All of the animals were handled according to approved institutional animal care and use committee (IACUC) protocols (#3661) of the University of Virginia. All surgery was performed under tricain anesthesia and every effort was made to minimize suffering.

## Decision letter and Author response

Decision letter  https://doi.org/10.7554/eLife.45241.040
Author response  https://doi.org/10.7554/eLife.45241.041

# Additional files

## Supplementary files

• Supplementary file 1. Transcriptome of DFCs at 80% epiboly in Control (Ctrl), Vgll4l and Yap/Taz loss-of-function.
DOI: https://doi.org/10.7554/eLife.45241.027

• Supplementary file 2. Regulation of the expression by Vgll4l and Yap/Taz of zebrafish homologs of Yap direct target genes in mammals. Table summarizing the variation of expression (fold change) of genes differentially expressed (normalized counts > 1, |log2foldchange| $\geq$ 1, adjusted P value $\leq$ 0,05) between control and Vgll4l or Yap/Taz morphants for 143 zebrafish homologs of Yap direct target genes in mammals. Reference source for the set of Yap direct target genes: (1) (*Zanconato et al., 2015*), (2) (*Wang et al., 2018*), (3) (*Lin et al., 2015*)
DOI: https://doi.org/10.7554/eLife.45241.028

• Supplementary file 3. Expression of genes known to be required for DFCs and/or KV development in control and in Vgll4l or Yap/Taz loss-of-function condition.
DOI: https://doi.org/10.7554/eLife.45241.029

• Supplementary file 4. Expression of genes coding for proteins involved in ciliogenesis and known to be required for proper function of the LRO in control and in Vgll4l or Yap/Taz loss-of-function condition.
DOI: https://doi.org/10.7554/eLife.45241.030

• Supplementary file 5. Sequence of primers used to generate sgRNAs and for screening Crispr/Cas9 mutants.
DOI: https://doi.org/10.7554/eLife.45241.031

• Supplementary file 6. Position of MO, ASO target sequences and of mutations in *vgll4l*, *vgll4b*, *yap*, *taz*, *tead1a* and *tead3a*.
DOI: https://doi.org/10.7554/eLife.45241.032

• Supplementary file 7. Sequences of mutations in *vgll4l*, *vgll4b*, *yap*, *taz*, *tead1a* and *tead3a*.
DOI: https://doi.org/10.7554/eLife.45241.033

• Supplementary file 8. Position of MO target sequences and mutations in *dnmt3bb.1*, *dnmt3bb.2*, *dnmt3ba*, *mbd3a* and *mbd3b*.
DOI: https://doi.org/10.7554/eLife.45241.034

• Supplementary file 9. Sequence of *dnmt3bb.1*, *dnmt3bb.2* and *dnmt3ba* mutants.
DOI: https://doi.org/10.7554/eLife.45241.035

• Transparent reporting form
DOI: https://doi.org/10.7554/eLife.45241.036

## Data availability

RNA sequencing data that support the findings of this study have been deposited in the Gene Expression Omnibus (GEO) under accession code GSE119623 and are also provided in

Supplementary file 1. All data generated or analysed during this study are included in the manuscript and supporting files. Source data for Figure 1, 2, 3, 9, Figure 1—figure supplement 2 and Figure 2—figure supplement 2 has been provided.

The following dataset was generated:

| Author(s) | Year | Dataset title | Dataset URL | Database and Identifier |
|---|---|---|---|---|
| Fillatre J, Thisse C, Thisse B | 2018 | RNA-seq of zebrafish embryo dorsal forerunner cells lacking Vestigial like 4 like (Vgll4l) or Yes associated protein 1 (Yap1)/ WW domain containing transcription regulator 1 (Wwtr1/Taz) activities | https://www.ncbi.nlm.nih.gov/geo/query/acc.cgi?acc=GSE119623 | NCBI Gene Expression Omnibus, GSE119623 |

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
