## [Decision Letter]

Thank you for sending your article entitled "Transcription factors downstream of hippo signaling control the establishment of left-right asymmetry in zebrafish" for peer review at *eLife*. Your article is being evaluated by Didier Stainier as the Senior Editor, a guest Reviewing Editor, and three reviewers.

All three reviewers were enthusiastic about many aspects of your manuscript but also asked for rather detailed changes that are likely to take considerable time. This includes improving several points of the study, in particular the part showing that VGLL4 acts in parallel of YAP/TAZ and clarification about the activity of YAP/TAZ during KV morphogenesis. The reviewers also felt that the gene list directly or indirectly regulated by hippo transcription factors provided did not help to understand better what is the function of hippo signaling in KV development.

*Reviewer #1:*

Summary:

In the manuscript "Transcription factors downstream of hippo signaling control the establishment of left-right asymmetry in zebrafish", Fillatre et al., investigate whether transcriptional components of the Hippo pathway regulate the formation of the Left-Right Organizer. Using genetic knockdowns/knockouts and pharmacological manipulations, the authors report some interesting phenotypes that suggest important roles of VGLL4 and YAP/TAZ for the left-right organization in the developing zebrafish embryo. Especially, the role in cilia formation, as well as the potential role of VGLL4 in controlling epigenetic gene expression, are of interest. However, the paper is still preliminary and suffers from the reliance on just loss-of-function approaches, whose specificity have not been sufficiently validated. Another weak point is the correlative nature of most of the studies. As illustrated, it remains unclear if and how VGLL4 regulates YAP/TAZ signaling, and whether changes in this signaling module are directly linked to the formation of the Left-Right Organizer. Thus, the mechanistic aspect of this paper is weak.

Essential revisions:

1) The authors use genetic and pharmacological approaches that interfere with YAP/TAZ-TEAD- and VGLL-dependent transcription in all cells and tissues. Given the essentiality of this transcriptional mechanism for general growth and development, one wonders whether the laterality defects are secondary to changes in tissue morphogenesis.

2) The authors do not provide direct evidence that VGLL4 interferes with YAP/TAZ-TEAD function. Thus, it remains unclear whether VGLL4 signals through its canonical pathway or via alternative transcription factors. What happens to prototypical YAP/TAZ target genes in LRO progenitors sorted form Tg(sox17:GFP)S870 cells? This should give clues on how VGLL4 functions. As well what about the expression of YAP/TAZ and their targets in YAP/TAZ knockdown/knockout cells?

3) It is not clear from the data shown whether YAP/TAZ and VGLL4 have a specific role in the formation of the Kupffer vesicle (KV) or whether their loss has general effects on dorsal forerunner cell (DFC) proliferation and viability. Effects on the latter might explain the vast changes in gene expression observed in the different mutants. Are DFC-specific genes direct YAP/TAZ target genes that are co-regulated by VGLL4?

4) The authors show that components of Hippo signaling are necessary for KV formation but what about sufficiency? Does activation or nuclear localization of YAP or TAZ drive DFC expansion?

*Reviewer #2:*

In this manuscript, Fillatre et al., present an interesting study of the effects of Hippo signalling pathway on the establishment of left-right asymmetry in zebrafish. Using different genetic tools, the authors show that transcription factors and cofactors that are downstream effectors of the Hippo pathway are necessary for the formation of LRO. They also perform a transcriptomic analysis on DFCs of Vgll4l and yap/taz morphants to show that these factors affect a number of genes previously shown to be expressed in DFCs, a panel of signalling pathways and finally different readers and writers that affect the epigenetic landscape of these cells. All in all, the study is interesting and thorough and the experiments performed were carefully designed and executed. One major problem is that this study does not permit the assessment of the genes that are directly regulated by these transcription factors and co-factors.

Of course, it is difficult to perform experiments such as chromatin immunoprecipitation in zebrafish, but the authors could provide some more insights regarding this aspect:

1) The authors could perform chromatin accessibility assays and check for motifs of the Hippo pathway. This experiment is doable with low cell numbers and could give a hint of the genes that are directly regulated by the Hippo pathway.

2) The authors perform their transcriptomic analysis on morphants. In this way the targeted gene is affected in the whole animal from the beginning of development and this could lead to many secondary effects. Since the authors have performed some of their genetic experiments on DFC specific knockdowns, maybe they can check by qPCR whether deregulated genes identified in their transcriptomic analysis are also deregulated in these tissue specific knockdown animals. In this way they could identify a tighter list of genes that are regulated by Hippo signalling transcription factors.

3) The authors performed a partial rescue experiment by expressing dnmt3bb.1. This rescue also provides an opportunity to check by transcriptomic analysis what is the expression profile during this rescue and provide a list of genes that are necessary for DFC formation.

*Reviewer #3:*

There are two major concerns with the present study:

1) It is not sufficiently clear to what degree the different transcriptional effectors of the Hippo signaling pathway differ in their function during KV development. Wouldn't one expect, for instance, that l-o-f of yap/taz has the opposite effect than that of vgll? Along the same vein, it would be important to see how the subcellular/nuclear localization of the analyzed transcription factors is dynamically regulated in DFCs and/or KV cells. For YAP and TAZ, for instance, there are previously characterized antibodies available that could be used. This might provide important additional insight into their expected functions in this process.

2) The authors identify a very large number (8-9k) of genes, the expression which is controlled by vgll and/or yap/taz. That said, it is perhaps not entirely surprising to find within those targets genes previously involved in KV development. It is, however, quite worrying to see that a large fraction (10/24) of the genes the authors have shown to be strongly regulated by in situ hybridization, were not amongst those targets. In the light of this, it is not clear how meaningful the transcriptomic approach has been and what can be learned from it.

---

## [Author Response]

Reviewer #1:

Summary:In the manuscript "Transcription factors downstream of hippo signaling control the establishment of left-right asymmetry in zebrafish", Fillatre et al., investigate whether transcriptional components of the Hippo pathway regulate the formation of the Left-Right Organizer. Using genetic knockdowns/knockouts and pharmacological manipulations, the authors report some interesting phenotypes that suggest important roles of VGLL4 and YAP/TAZ for the left-right organization in the developing zebrafish embryo. Especially, the role in cilia formation, as well as the potential role of VGLL4 in controlling epigenetic gene expression, are of interest. However, the paper is still preliminary and suffers from the reliance on just loss-of-function approaches, whose specificity have not been sufficiently validated.

This is correct that our analysis is mainly based on loss of function experiments. They demonstrate the requirement for each of the 6 genes analyzed in the formation of the zebrafish left-right organizer (LRO). In particular we found that the activity of each of these 6 transcription factors and cofactors are needed in DFCs for the formation and function of the Kupffer’s vesicle and for the assembly and function of motile cilia that are essential to the activity of the LRO. Therefore, loss-of-function approaches are particularly appropriate to demonstrate the requirement of these transcription factors and cofactors to the proper differentiation of the LRO progenitors into a functional Kupffer’s vesicle.

We are not sure we understand the reviewer’s comments about the lack of validation of these loss-of-function experiments. In this study, we used a combination of strategies to generate loss-of-function conditions including: knockdowns, pharmacological drug treatments and genetic approaches to establish the requirement of each of the 6 transcription factors and cofactors: Vgll4l, Vgll4b, Yap, Taz, Tead1a and Tead3a, in the formation of the zebrafish LR organizer (e.g., Figure 1, Figure 1—figure supplement 2).

Results from knockdown experiments have been validated by:

1) Using different, non-overlapping, morpholinos (MO) or antisense oligonucleotides (ASO) and showing they result in the same loss of function phenotype.

2) By targeting either translation or splicing and obtaining the defects.

3) By rescuing knockdown phenotypes through injections of morpholinos insensitive in vitro synthesized mRNAs.

4) For Yap we also used a specific pharmacological drug and observed the same phenotypes as those observed for MO knockdown of Yap function on the formation of the LRO.

5) We confirmed data obtained in knockdown experiments by generating Crispr-Cas9 mutants that display identical phenotypic defects. This is shown in a new figure (Figure 1—figure supplement 3). In this figure we also show that the global morphology of homozygous mutant embryos is not affected indicating that the effect on DFCs/KV is not a secondary consequence of a global defect of embryo development.

In addition, in the revised version of our manuscript we provide phenotypic analysis of KV defects in homozygous mutant embryos (Figure 2—figure supplement 1).

6) In addition to loss of function studies, we have performed DFC specific gain of function of Yap presented in (Figure 2—figure supplement 2) that shows that as known for Yap function in other vertebrates, increase of its activity results in an increased size of the organ associated with an increased number of cells.

7) Finally, we took advantage of a unique property of DFCs: these cells are the last embryonic cells to keep cytoplasmic connections with the yolk syncytial layer. Therefore, they can be specifically targeted by MO injection into the yolk after the other embryonic cells have lost their connection to the yolk syncytial layer. This allows DFCs specific knockdown (as well as DFC specific gain of function). This technique has been established by Amack and Yost, 2004 and is widely used by the community working on DFCs and Kupffer’s vesicle. A detailed protocol has been published in Wang, Yost and Amack, 2013. Using this approach, we have been able to examine the tissue specific effect of the loss of function of each gene in DFCs without affecting other embryonic cells.

Altogether, all the experimental approaches listed above, the controls we performed and the identity of phenotype on the formation of the LRO we observed in each experimental condition provide a full validation that lack of any of the 6 genes analyzed results in disruption of the differentiation of DFCs into the functional zebrafish LRO.

Another weak point is the correlative nature of most of the studies. As illustrated, it remains unclear if and how VGLL4 regulates YAP/TAZ signaling, and whether changes in this signaling module are directly linked to the formation of the Left-Right Organizer. Thus, the mechanistic aspect of this paper is weak.

Our comments about the second major criticism of reviewer 1 (about the regulation of Yap/Taz by Vgll4) is detailed below in our responses to his/her different major comments.

Essential revisions:1) The authors use genetic and pharmacological approaches that interfere with YAP/TAZ-TEAD- and VGLL-dependent transcription in all cells and tissues. Given the essentiality of this transcriptional mechanism for general growth and development, one wonders whether the laterality defects are secondary to changes in tissue morphogenesis.

We understand the concern of reviewer 1 about a possible secondary effect of Yap/TazTEAD and Vgll4 on DFCs due to general changes in tissue morphogenesis. However, two lines of evidence show that impact of the lack of any of the transcription factors and cofactors on KV formation is not a secondary consequence of a global change in growth and development of the embryo:

1) Despite of their essential role in controlling general growth and development, individual mutants of Yap1, Taz, Vgll4l, Vgll4b and Tead3a display a normal morphology at late embryonic stages and are able to give rise to fertile adults (Figure 1—figure supplement 3). This observation has been previously reported for Yap and for Taz (Miesfield et al., 2015). Therefore, we observe strong defects in the formation of the LRO (and in consequence on the LR asymmetry of the embryo) in both mutants and morphants while the global morphology, growth and development of the embryos is unaffected. This shows that the lack of function of each of these different genes is not likely to affect the formation of the Kupffer’s vesicle as a secondary consequence of a global change in embryonic development and growth.

We provide two additional figures to illustrate this point:

1) in Figure1—figure supplement 3 we show that the homozygous mutant embryos at two days of development (a time where the LR asymmetry is already established) do not have major developmental defects while (2) their Kupffer’s vesicle (observed at early somitogensis) is strongly affected (Figure 2—figure supplement 1).

2) As stated above, we found the same strong defects in LRO formation and in the laterality of the embryo in DFCs specific knockdown experiments (as exemplified in Figure1—figure supplement 2). In these conditions, defects observed in formation of the Kupffer’s vesicle result only from specific knockdowns of each gene in the LRO progenitors. Therefore, the phenotype we observe cannot be an indirect secondary consequence of an early global effect on the embryonic development.

2) The authors do not provide direct evidence that VGLL4 interferes with YAP/TAZ-TEAD function. Thus, it remains unclear whether VGLL4 signals through its canonical pathway or via alternative transcription factors.

This is correct, we do not provide evidence of interference of Vgll4 with Yap/Taz-TEAD because we didn’t find any evidence for such interference in DFCs during their differentiation into the Kupffer’s vesicle. Based on published data we initially hypothesized that Vgll4s and Yap/Taz may have opposite effects and that Vgll4s will interfere with Yap/Taz as it has been shown in other model systems. Our observations however do not support this mechanism in the DFCs during the formation of the LRO. Here are the pieces of evidence:

1) The Kupffer’s vesicle phenotype is very similar between Vgll4s and Yap/Taz loss of function, while an antagonistic effect of Vgll4s and Yap/Taz should have generated different/opposite phenotypes.

2) The Kupffer’s vesicle phenotype of *tead1a* and *tead3a* is also similar to both Vgll4s and Yap/Taz loss of function.

3) Transcriptome data show that for genes regulated by both Vgll4l and Yap/Taz, a vast majority (84%) is regulated similarly by Vgll4l and Yap/Taz while only 16% are regulated in opposite way (see Venn diagram Figure 5B).

What happens to prototypical YAP/TAZ target genes in LRO progenitors sorted form Tg(sox17:GFP)S870 cells? This should give clues on how VGLL4 functions.

We analyzed the impact of Yap/Taz and Vgll4 loss of function on the transcription of Yap direct target genes identified in other systems.

We performed a compilation of 380 mammalian genes (human and rat) from 3 publications: Zanconato et al., 2015; Wang et al., 2018; Lin, Z et al., 2015.

In the zebrafish genome (GRCz10) we found 318 homologues for these 380 Yap direct target genes. Looking at our transcriptome data we found that 143 out of the 318 homologues were expressed in the DFCs (normalized counts ≥1 in any of these conditions: control, vgll4l Yap, Taz, Tead1a and Tead3a, in the formation of the zebrafish LR organizer (e.g.: Figure 1, Figure 1—figure supplement 2). (llog2foldchangel≥1 and a justed P value ≤ 0.05).

This set of 143 genes correspond to the zebrafish homologues of direct targets of Yap (likely to be also direct targets in zebrafish) whose expression is regulated by both Yap/Taz and Vgll4l in the DFCs. If Vgll4l acts as an antagonist of Yap/Taz by competing for binding of Teads we should expect to see opposite regulation of these genes in Vgll4l and Yap/Taz loss of function.

This is not what we observe: we found that 118/143 genes (82.5%) were regulated similarly in both conditions (61 downregulated and 57 upregulated in both Vgll4l and Yap/Taz loss of function) and only 25/143 (17.5%) were regulated in opposite ways. This strongly supports that Vgll4l and Yap/Taz do not act antagonistically during the process of LRO formation.

We now discuss these data in the text and provide a table (Supplementary file 2) with the transcriptome data for these 143 genes.

As well what about the expression of YAP/TAZ and their targets in YAP/TAZ knockdown/knockout cells?

The expression of Yap/Taz and their targets in knockdown cells is provided in the transcriptome (Supplement file 1). The expression of 143 direct targets of Yap are provided together with Vgll4l data in Supplement file 2.

For the expression of the main Yap target genes reported so far (mainly in human cancer cells), a list is provided below: with the number of normalized counts for each gene in control and in Yap/Taz morphant as well as Log2foldchange for expression of Yap direct genes and the references for studies identifying these genes as direct Yap targets.

The analysis of the expression of the zebrafish homologue of 39 Yap direct targets identified in mammals (pulled out from 10 different studies) reveals that 15 are downregulated in Yap/Taz morphants indicating that Yap/Taz act as activators of these genes; 7 genes are upregulated indicating that Yap/Taz act as repressors of their expression; finally 17 genes are not expressed and/or not regulated by Yap/Taz in DFCs.

While *ddit4* has been shown to be repressed by Yap/Taz in mammal (Kim et al., 2015), all the other Yap targets have been described to be activated.

We found that only 16 genes (15 activated genes plus *ddit4*) in this list of 39 genes are regulated similarly in KV progenitors and in mammalian cells.

These differences in gene regulation (essentially between cancer and embryonic cells during organogenesis) indicate that Yap controls different sets of genes in zebrafish during organogenesis and in human cancer cells. Because this observation does not provide any information about the role of Yap in the formation of the LRO we decided not to add the table presented in Author response table 1 in the manuscript.

**Author response table 1. resptable1:** 

	**Gene**	**Ctrl**	**Yap/Taz**	**Yap/Taz FC**	**ref**	
1	igfbp3	278	**0**	**-9.8**	6	**activated by Yap/Taz in DFCs**
2	F3	127	**0**	**-9.4**	6	
3	crim1	60	**0**	**-8.3**	6	
4	Diaph3	57	**0**	**-7.6**	9	
5	rbms3	47	**0**	**-7.4**	6	
6	Cdc42ep3	21	**0**	**-6.9**	9	
7	ccdc80	27	**0**	**-6.6**	6	
8	nuak2	251	**16**	**-3.9**	6	
9	ptgs2a	110	**11**	**-3.2**	4	
10	anln	908	**182**	**-2.3**	9	
11	gpatch4	262	**73**	**-1.8**	10	
12	lmnb2	3219	**1385**	**-1.2**	10	
13	amotl2a	912	**394**	**-1.2**	6	
14	dock5	11	**4**	**-1.2**	6	
15	myof	1037	**428**	**-1.2**	6	
1	ddit4	1235	**2830**	**1.1**	8	**repressed by Yap/Taz in DFCs**
2	txn	767	**4574**	**2.5**	5	
3	Thbs1	2	**17**	**2.8**	9	
4	ankrd1a	0	**59**	**6.8**	1	
5	ankrd1a	0	**59**	**6.8**	6	
6	gadd45ab	1	**202**	**7**	6	
7	Plaua	0	**77**	**7.3**	7	
8	Serpine1	0	201	**8.6**	9	
1	ankrd1b	0	0	NS	1	**not regulated by Yap/Taz in DFCs**
2	cat	106	88	NS	2	
3	ptgs2b	0	0	NS	4	
4	wsb2	0	0	NS	10	
5	cyr61	0	0	NS	6	
6	ctgfa	0	0	NS	6	
7	ctgfb	0	0	NS	6	
8	amotl2b	21	12	NS	6	
9	ankrd1b	0	0	NS	6	
10	last2	157	267	NS	6	
11	gadd45aa	0	0	NS	6	
12	tgfb2	0	0	NS	6	
13	nt5e	0	0	NS	6	
14	foxf2	0	0	NS	6	
15	asap1	0	0	NS	6	
16	Plaub	0	0	NS	7	
17	Wtip	0	1	NS	9	

3) It is not clear from the data shown whether YAP/TAZ and VGLL4 have a specific role in the formation of the Kupffer vesicle (KV) or whether their loss has general effects on dorsal forerunner cell (DFC) proliferation and viability. Effects on the latter might explain the vast changes in gene expression observed in the different mutants.

There is an effect on proliferation and viability of DFCs but most of these cells do not die. A vast majority of DFCs used for the transcriptome analysis were not apoptotic (in average 1 or 2 apoptotic cells out of ~25-38 cells present in DFC clusters analyzed by RNA-seq). In addition, we found that lack of Vgll4l and Yap/Taz functions affects the differentiation of these cells. In particular they strongly affect genes involved in cilia organization and cilia motility. Yap has been reported to be involved in formation of non-motile cilia during zebrafish kidney development (He et al., 2015). This suggests that in addition to its well-known control of organ growth, Yap plays also a specific role in ciliogenesis. Our observation of the strong disruption of motile ciliogenesis in Yap/Taz morphants (Figure 3) supports the specific role of these TcoFs in the regulation of ciliogenesis.

Finally, in addition of a large set of genes regulated by both Yap/Taz and Vgll4l we found genes regulated only by Yap/Taz or only by Vgll4l. For example, 759 genes are downregulated only in Yap/Taz morphants while 1209 other genes are downregulated only in Vgll4l loss of function (Figure 5B). If the effect of Yap/Taz and Vgll4l on the formation of the LRO was simply resulting from a global disruption of cell physiology when their viability is impaired the transcriptome of DFCs in Yap/Taz and Vgll4l morphants would appear more similar. In particular, GO term analysis (Figure 5C) shows that both Yap/Taz and Vgll4l affect cilium movement, cilium organization and epithelium development but that Yap/Taz regulates mitotic cell cycle process (in agreement with the known function of these transcription cofactors) while Vgll4l regulates genes involved in covalent chromatin modification. Therefore, while the impact of Yap/Taz and Vgll4l on the DFC transcriptome is wide, their effects are specific, ruling out that the change in gene expression observed in the different morphants is simply a secondary consequence of an impairment in cell viability.

Are DFC-specific genes direct YAP/TAZ target genes that are co-regulated by VGLL4?

Direct Yap/Taz targets in DFCs are unknown. Due to the very small number of DFCs (50 cells in WT) we cannot expect to be able to perform ChIP-seq experiments comparable to the quality published for human cells in culture (with millions of cells used).

As we proposed we performed ChIP-seq to try to characterize direct target genes for Vgll4l and Yap. For this experiment we used cells of the whole embryo isolated at late gastrula stage after injection of in the egg of mRNAs coding an Active-motif tagged versions of Yap or Vgll4l. Despite several attempts with 300 embryos injected per condition the amount of DNA was very low and as a result the number of specific ChIP-seq peaks was low as well. The final number of binding sites identified was too small to allow a statistical analysis of the regulation of Yap by Vgll4l on their direct target genes. Even with a good number of peaks, the result would have only been indicative as the analysis was done on the whole embryo and not on DFCs (which represent only 1% of the total number of cells at midgastrula stage).

4) The authors show that components of Hippo signaling are necessary for KV formation but what about sufficiency? Does activation or nuclear localization of YAP or TAZ drive DFC expansion?

Neither Vgll4 nor Yap/Taz are sufficient to induce formation of the LRO. As we stated in the text; “in situ hybridization for sox17, an early DFC marker, we found that DFC clusters are present at early gastrula stage for every Hippo TFs/TcoFs mutant/morphant tested” and the number of DFCs at early gastrula stage (Figure 2F) is the same for both WT and morphant embryos”. These TFs and TcoFs are not involved in the specification of the KV progenitors but are required at gastrula stage during the process of their differentiation into the KV.

The KV progenitors, the DFCs, are specified at late blastula stage in response to Nodal signaling. Therefore, while constitutively activated Activin receptor I (Acvr1b, the Nodal receptor) can induce DFCs at the animal pole of a zebrafish blastula (Compagnon et al., 2014) gain of functions of Taz, Yap or of a constitutive version of Yap (Yap5SA) are not sufficient to induce formation of ectopic DFCs. Of course, gain of function of Yap protein (known to stimulate proliferation) by injection of in vitro synthesized Yap mRNA in a DFC specific manner (that is after other cells of the embryo lose their cytoplasmic bridges with the yolk syncytial layer) results in an increase of the Kupffer’s vesicle size associated with an increased number of cells in the enlarged Kupffer’s vesicle. This experiment has been completed and the results are presented in Figure 2—figure supplement 2.

Reviewer #2:

[…] One major problem is that this study does not permit the assessment of the genes that are directly regulated by these transcription factors and co-factors.Of course, it is difficult to perform experiments such as chromatin immunoprecipitation in zebrafish, but the authors could provide some more insights regarding this aspect:1) The authors could perform chromatin accessibility assays and check for motifs of the Hippo pathway. This experiment is doable with low cell numbers and could give a hint of the genes that are directly regulated by the Hippo pathway.

We agree that ATAC-seq may be doable with a reasonable number of cells. Performing this experiment on DFCs will show the open chromatin sites (we can anticipate a very large number). Then we will have to check for possible TEAD1a or TEAD3a motifs in open chromatin domains. TEAD binding motif has not been characterized using zebrafish proteins and we will have to use vertebrate TEAD consensus sequences. Based on their frequency on the genome sequence we anticipate to have a quite large number of candidates.

As an alternative to ATAC-seq we proposed to perform ChIP-seq to try to characterize direct target genes for Vgll4l and Yap. Because of the small number of DFCs, for this experiment we used cells of the whole embryo isolated at late gastrula stage after injection of in the egg of mRNAs coding an Active-motif tagged versions of Yap or Vgll4l. Despite several attempts with 300 embryos injected per condition the amount of DNA was very low and as a result the number of specific ChIP-seq peaks was low as well. The final number of binding sites identified was too small to allow a statistical analysis of the regulation of Yap by Vgll4l on their direct target genes. Even with a good number of peaks, the result would have only been indicative as the analysis was done on the whole embryo and not on DFC cells (which represent only 1% of the total number of cells at midgastrula stage).

2) The authors perform their transcriptomic analysis on morphants. In this way the targeted gene is affected in the whole animal from the beginning of development and this could lead to many secondary effects. Since the authors have performed some of their genetic experiments on DFC specific knockdowns, maybe they can check by qPCR whether deregulated genes identified in their transcriptomic analysis are also deregulated in these tissue specific knockdown animals. In this way they could identify a tighter list of genes that are regulated by Hippo signalling transcription factors.

We performed in situ hybridization on DFC specific knockdowns for some of the DFC genes (Figure 4) we found downregulated in global knockdowns. Results are presented in Figure 4—figure supplement 1. For all analyzed genes we observed the same reduction of expression in global knockdowns and in DFC specific knockdowns. In addition, we should mention that while global knockdowns affect the activity of the different TFs and TcoFs in the whole animal, we didn’t observe any effect on the morphology of the embryo. This is shown in Figure1—figure supplement 3: homozygous mutant embryos do not display morphological defects (and most of them reach adulthood) while their laterality is strongly affected. Because other tissues and organs are formed normally and are fully functional, this strongly suggests that the effect observed for the loss of function of these TFs and TcoFs on the differentiation of the progenitors of the LRO is not the secondary consequence of an earlier broad effect of their loss of function on the development of the embryo.

3) The authors performed a partial rescue experiment by expressing dnmt3bb.1. This rescue also provides an opportunity to check by transcriptomic analysis what is the expression profile during this rescue and provide a list of genes that are necessary for DFC formation.

We agree with reviewer 2 about the interest of this analysis. We are planning to perform in the future an extensive analysis of the impact of readers and writers of methylation marks on the transcriptome of DFCs but we think this interesting question deserves a full study, which is beyond the scope of the current manuscript.

Reviewer #3:

There are two major concerns with the present study:1) It is not sufficiently clear to what degree the different transcriptional effectors of the Hippo signaling pathway differ in their function during KV development. Wouldn't one expect, for instance, that l-o-f of yap/taz has the opposite effect than that of vgll?

An opposite effect of Yap/Taz and Vgll4l loss of function is what we initially expected, based on the current model of interaction between these TcoFs in control of organs growth or in a cancer cells model system. However, our study of the impact of the lack of these genes function on the transcriptome of the DFCs supports that this is not the case during the formation of the LRO:

Loss of function phenotypes (mutant, morphant, drug, antisense oligonucleotides) are similar for all genes. Most of our data support that Yap/Taz and Vgll4 have a similar role in the regulation of the number of DFCs at the end of gastrulation and in the formation of the ciliated epithelium of the LRO.

Most of differentially expressed genes identified in our transcriptome analysis are regulated similarly by Vgll4l and Yap/Taz and this is also true for the zebrafish homologues of direct Yap targets in mammals.

Along the same vein, it would be important to see how the subcellular/nuclear localization of the analyzed transcription factors is dynamically regulated in DFCs and/or KV cells. For YAP and TAZ, for instance, there are previously characterized antibodies available that could be used. This might provide important additional insight into their expected functions in this process.

An opposite effect of Yap/Taz and Vgll4l loss of function is what we initially expected, based on the current model of interaction between these TcoFs in control of organs growth or in a cancer cells model system. However, our study of the impact of the lack of these genes function on the transcriptome of the DFCs supports that this is not the case during the formation of the LRO:

Loss of function phenotypes (mutant, morphant, drug, antisense oligonucleotides) are similar for all genes. Most of our data support that Yap/Taz and Vgll4 have a similar role in the regulation of the number of DFCs at the end of gastrulation and in the formation of the ciliated epithelium of the LRO.

Most of differentially expressed genes identified in our transcriptome analysis are regulated similarly by Vgll4l and Yap/Taz and this is also true for the zebrafish homologues of direct Yap targets in mammals.

2) The authors identify a very large number (8-9k) of genes, the expression which is controlled by vgll and/or yap/taz. That said, it is perhaps not entirely surprising to find within those targets genes previously involved in KV development. It is, however, quite worrying to see that a large fraction (10/24) of the genes the authors have shown to be strongly regulated by in situ hybridization, were not amongst those targets. In the light of this, it is not clear how meaningful the transcriptomic approach has been and what can be learned from it.

An opposite effect of Yap/Taz and Vgll4l loss of function is what we initially expected, based on the current model of interaction between these TcoFs in control of organs growth or in a cancer cells model system. However, our study of the impact of the lack of these genes function on the transcriptome of the DFCs supports that this is not the case during the formation of the LRO:

Loss of function phenotypes (mutant, morphant, drug, antisense oligonucleotides) are similar for all genes. Most of our data support that Yap/Taz and Vgll4 have a similar role in the regulation of the number of DFCs at the end of gastrulation and in the formation of the ciliated epithelium of the LRO.

Most of differentially expressed genes identified in our transcriptome analysis are regulated similarly by Vgll4l and Yap/Taz and this is also true for the zebrafish homologues of direct Yap targets in mammals.

Author response table 1 reference list

1- Choi HJ, Zhang H, Park H, Choi KS, Lee HW, Agrawal V, et al. Yes-associated protein regulates endothelial cell contact-mediated expression of angiopoietin-2. Nature communications. 2015;6:6943.

2 -Shao D, Zhai P, Del Re DP, Sciarretta S, Yabuta N, Nojima H, et al. A functional interaction between Hippo-YAP signalling and FoxO1 mediates the oxidative stress response. Nature communications. 2014;5:3315.

3 - Zhao B, Ye X, Yu J, Li L, Li W, Li S, et al. TEAD mediates YAP-dependent gene induction and growth control. Genes Dev. 2008;22:1962-71.

4- Basu-Roy U, Bayin NS, Rattanakorn K, Han E, Placantonakis DG, Mansukhani A, et al. Sox2 antagonizes the Hippo pathway to maintain stemness in cancer cells. Nature communications. 2015;6:6411.

5- Yuan T, Rafizadeh S, Azizi Z, Lupse B, Gorrepati KD, Awal S, et al. Proproliferative and antiapoptotic action of exogenously introduced YAP in pancreatic beta cells. JCI insight. 2016;1:e86326.

6 - Wang et al., 2018, Comprehensive Molecular Characterization of the Hippo Signaling Pathway in Cancer. Cell Reports 25, 1304–1317.

7 - Corley SM, Mendoza-Reinoso V, Giles N, Singer ES, Common JE, Wilkins MR, Beverdam A. Plau and Tgfbr3 are YAP-regulated genes that promote keratinocyte proliferation. Cell Death Dis. 2018 Oct 31;9(11):1106. doi: 10.1038/s41419-018-1141-5.

8 - Minchul Kim, Taekhoon Kim, Randy L. Johnson, Dae-Sik Lim Transcriptional Co-repressor Function of the Hippo Pathway Transducers YAP and TAZ Cell Rep. 2015 Apr 14;11(2):270-82. doi: 10.1016/j.celrep.2015.03.015. Epub 2015 Apr 2.

9 - Foster CT, Gualdrini F, Treisman R. Mutual dependence of the MRTF-SRF and YAP-TEAD pathways in cancer-associated fibroblasts is indirect and mediated by cytoskeletal dynamics. Genes Dev. 2017 Dec 1;31(23-24):2361-2375.

10 - Zanconato F, Forcato M, Battilana G, Azzolin L, Quaranta E, Bodega B, et al. Genome-wide association between YAP/TAZ/TEAD and AP-1 at enhancers drives oncogenic growth. Nat Cell Biol. 2015;17:1218-27.